# Immunogenicity of two representative American consensus scorpion neurotoxins from the genera *Tityus* and *Centruroides*

Samuel Cardoso-Arenas[1], Miguel Angel Mejia-Sanchez[1], Ricardo Miranda-Blancas[1], Herlinda Clement[1], Lilu Corrales-García[1,2], Ivan Arenas[1], Gerardo Pavel Espino-Solís[3], Hildaura Acosta[4], Marcos H. Salazar[4,5], Gerardo Corzo[1]*

1 Departamento de Medicina Molecular y Bioprocesos, Instituto de Biotecnología, Universidad Nacional Autónoma de México, Cuernavaca, Morelos, México, 2 PECET—Programa para el Estudio y Control de Enfermedades Tropicales, Facultad de Medicina, Universidad de Antioquia, Medellín, Colombia, 3 Facultad de Medicina y Ciencias Biomédicas, Universidad Autónoma de Chihuahua, Circuito Universitario s/n, Campus II, Chihuahua, México, 4 Universidad de Panamá, Facultad de Medicina, Centro de Investigación e Información de Medicamentos y Tóxicos, Ciudad de Panamá, Panamá, 5 Universidad de Panamá, Facultad de Ciencias Naturales, Exactas y Tecnología, Departamento de Bioquímica, Ciudad de Panamá, Panamá

* gerardo.corzo@ibt.unam.mx

## Abstract

Two consensus scorpion toxins derived from venoms of the genera *Centruroides* (NATx, North America), and *Tityus* (SATx, South America) were rationally designed and recombinantly expressed in *Escherichia coli* Origami. Both recombinant proteins were produced as inclusion bodies and subsequently purified and refolded *in vitro* to obtain biologically active isoforms. The expression yields were approximately 1 mg/L for rNATx and 0.5 mg/L for rSATx. Structural characterization by circular dichroism spectroscopy revealed that rNATx and rSATx exhibited folding patterns typical of scorpion β-toxins. The purified recombinant toxins were used as immunogens to raise polyclonal antibodies in New Zealand rabbits. The resulting antisera were evaluated for their capacity to neutralize isolated toxins and whole scorpion venoms. rNATx elicited a stronger immune response and showed superior immunogenicity compared to rSATx. Notably, 1-1.5 mg of anti-rNATx antibodies were sufficient to neutralize up to $3LD_{50}$ of venoms from the most medically relevant *Centruroides* species in México. On the other hand, approximately 20 mg of anti-rSATx antibodies were required to neutralize only $2LD_{50}$ of *Tityus* venoms. Additionally, T-cell subsets quantified by flow cytometry showed that rNATx is more immunogenic and probably confers improved antivenom efficacy, highlighting its potential application in the development of broad-spectrum antivenoms against scorpion envenomation.

**Data availability statement:** All data generated during this study are within the manuscript itself.

**Funding:** This work was funded by the "Dirección General de Asuntos del Personal Académico (DGAPA-UNAM)" grant number IT200724, and CONACyT/SECIHTI-PRONAII grant number 303045. SCA and MAMS are doctoral students from "Programa de Maestría y Doctorado en Ciencias Bioquímicas", they received fellowships CVU: 884453 and 836427, respectively, from CONACyT/SECIHTI. RMB acknowledges the scholarship granted by CONACyT/SECIHTI (CVU 592043) for his postdoctoral fellowship. The funders had no role in study design, data collection and analysis, decision to publish, or preparation of the manuscript.

**Competing interests:** The authors have declared that no competing interests exist.

## Author summary

Scorpion stings are considered a neglected health problem in tropical areas, with incidences reaching several hundred or thousands each year. Scorpion sting envenomation causes various symptoms, including pain, dizziness, nausea, vomiting, and in some cases, death. In the American continent, two genera of buthid scorpions are of medical importance: *Centruroides* and *Tityus*. Treating scorpion envenomations in American tropical countries involves the use of specific antivenoms; however, their efficacy may vary among scorpion species. In this work, we demonstrate that the *de novo* design of two scorpion toxins, here named rNATx and rSATx, which are representative of the genera *Centruroides* and *Tiyus*, respectively, was able to elicit an immune response in rabbits, whose antibodies were able to neutralize scorpion venoms and native toxins isolated from American buthid scorpions.

## 1 Introduction

The global incidence of scorpion envenomation is estimated to be up to 1.5 million accidents and 2,600 deaths per year [1,2]. The antivenom used nowadays is considered the unique treatment for scorpion envenoming [3]. Peptides are the most studied scorpion venom toxins because some of them are neurotoxic to humans. Although there are neurotoxins that act on different voltage-gated ion channels, such as $Na_v$, $K_v$, and $Ca_v$, those neurotoxins that affect $Na_v$ are considered the main components for triggering envenoming [4]. Neurotoxins that affect $Na_v$ can be separated into two groups despite sharing similar three-dimensional structures. The first group delays or changes the $Na_v$ inactivation, and the second one changes the opening of Nav to more negative potentials [5]. All these scorpion $Na^+$ toxins are peptides of 60–76 amino acids stabilized by four disulfide bonds [6]. That is, two segments of helix plus a triple-sheet are presented; these regions are connected by variable loops, adopting an α/β-structure [7].

In America, one of the two most dangerous genera of scorpions is represented by the *Tityus* genus, which is distributed in Central and South America [8]. The incidences by *Tityus* species could reach up to 30–200 cases/100,000 inhabitants in countries including mainly Trinidad and Tobago, Costa Rica, Panama, Colombia, Venezuela, Ecuador, Brazil, and Argentina. *T. serrulatus* and *T. bahiensis* have been the most studied venoms in Brazil; however, in recent years, other species that produce acute envenoming are *T. asthenes, T. championi, T. festae, T. jaimei, T. discrepans,* among others [9]. The geographic distribution of the *Tityus* species is diverse, and so is their venom. Only three antivenoms are available in South America, whose capability of neutralizing is focused on *T. serrulatus* (Brazil), *T. trivitatus* (Brazil, Argentina, and Paraguay), and *T. discrepans* (Venezuela) [9–13]. Therefore, it is challenging to neutralize all of them with antibodies from an antivenom raised from a single *Tityus* species venom.

On the other hand, the *Centruroides* species had a restricted distribution from the southern part of the USA, the Pacific area of Mexico, Central America, and some areas in Venezuela and Colombia [14]. In Mexico, scorpion stings are treated by a specific antivenom that neutralizes several venoms such as *C. limpidus, C. noxius, C. suffusus, C. tecomanus, C. infamatus, C. elegans,* and *C. sculpturatus* [15]. Yet, new reports have shown that some *Centruroides* species containing lethal mammalian toxins require specific single recombinant antibody fragments to be neutralized [16,17]. *Centruroides* antivenom is produced by Mexican companies that distribute to the USA, Mexico, and Central America [15–17].

Currently, there are no scorpion antivenoms that can successfully neutralize scorpion venoms from different genera. Consequently, strategies have been developed to produce better neutralizing antibodies against medically important scorpion stings. For example, specific toxins have been expressed through heterologous expression systems. Those recombinant toxins, usually obtained in *E. coli* systems, have been used to raise specific antibodies capable of neutralizing the lethal activity of native toxins and the whole scorpion venoms [18–23]. Recently, *in-silico* design of consensus animal toxins has been employed to create *de novo* recombinant antigens for experimental antivenoms to achieve broad-spectrum neutralization of regional or continental animal venoms [24,25].

Here, the design and heterologous expression of two consensus American scorpion toxins, rNATx and rSATx, produce antibodies with a broad spectrum against the venom of species from *Centruroides* and *Tityus*, respectively. So, these newly created neurotoxins could be used as immunogens instead of the whole scorpion venom for antibody production, or they could be used to enrich scorpion venoms when used as an immunogen.

## 2 Materials and methods

### 2.1 Ethical statement

No experiments with humans were performed. All applicable international, national, and/or institutional guidelines for the care and use of animals were followed, and procedures performed in the present study involving animals were done so by the bioethical standards at the "Instituto de Biotecnología - UNAM" (https://ibt.unam.mx/agrupacion/direccion-22/secretaria-academica-23/comite-de-bioetica-122). In addition, animal management was following the Animal Care guidelines from the Bioethics Committee currently used at the Instituto de Biotecnología, UNAM, which approved and authorized the animal work (Ethical approval CB/IBt/Project #451). Human red blood cells were collected from healthy donors who gave written consent for phlebotomy. This related experiment was approved by the Ethics Committee of the "*Facultad de Medicina y Ciencias Biomédicas*" of the Universidad Autónoma de Chihuahua under registration number CI-068–19.

LD$_{50}$ values were obtained from published sources and functionally tested *in vivo* using challenge doses consistent with reported lethality, without performing full LD$_{50}$ recalculations, in order to comply with institutional animal welfare guidelines.

All efforts were made to minimize animal suffering and to use the minimum number of animals necessary to obtain robust data.

### 2.2 Venoms and toxins

Venoms from *Tityus jaimei* and *T. festae* were kindly donated by CIIMET from the Universidad de Panamá. *T. serrulatus* venom was donated by Dr. Lourival Possani from the "Instituto de Biotecnología – UNAM", and all the *Centruroides* venoms were purchased through the Octolab (México). All the samples were dried and then dissolved in phosphate buffer (pH 7.0). They were centrifuged for 10 min at 17,600 *g*, and the soluble phase was transferred to a clean tube. The toxins Ts1 and Cn2 were isolated from the venoms as previously described [26].

### 2.3 Animals

CD-1 mice (18–20 g) and New Zealand rabbits (1.8-2 kg) were used and provided by the "Instituto de Biotecnología" Animal House. They were under controlled environments and under veterinary supervision. All the animals received food and

water *ad libitum.* Animal handling was always conducted to minimize suffering and discomfort, aiming to optimize animal welfare during research in accordance with Mexican legislation on the use of laboratory animals (Norma Oficial Mexicana, 1999, NOM-062-ZOO-1999).

## 2.4 Consensus sequence design

Neurotoxin primary structures were selected from the National Center for Biotechnology Information website (https://www.ncbi.nlm.nih.gov/pubmed) and the Uniprot protein database (http://www.uniprot.org). Once the structures were chosen, the neurotoxins with the lower median lethal dose and higher medical relevance were selected to create a consensus scorpion toxin by multiple sequence alignments using the Jalview software and the T-COFFEE online server. When amino acid positions with indeterminations existed, the most relevant amino acid was chosen according to its immunogenicity index established by Kolaskar and Tongaonkar [27].

## 2.5 Bacterial strains, plasmids, and enzymes

For cloning and vector amplification, *E. coli* XL1-Blue was used, and *E. coli* Origami was selected for protein expression of rNATx and rSATx. Plasmid pQE30 (Qiagen, CA, USA) was chosen for the cloning step of each recombinant toxin. The enzymes utilized in this communication were *Taq* polymerase, T4 DNA ligase, Factor Xa (FXa) as a protease, and restriction enzymes. All of them were acquired from New England Biolabs (NEB, MA, USA).

## 2.6 rNATx and rSATx vector construction

The NATx and SATx amino acid sequences were used to create their respective DNA sequences by reverse translation with the help of the Bioinformatics server (http://www.bioinformatics.org). After this step, each DNA sequence was optimized through the preferential codon usage of *E. coli* (http://www.kazusa.or.jp/codon). Four overlapping synthetic oligonucleotides were designed to link each recombinant consensus toxins. The respective oligonucleotides were synthesized by the "Unidad de Síntesis y Secuenciación de DNA" from the "Instituto de Biotecnología". Furthermore, some restriction sites were added, including *Bam*H1 and *Pst*I forward (rNATx-Up1) and reverse (rNATx-Lw4), respectively. Having the opportunity of removing the His-tag added by the vector used, a sequence for protease Factor Xa (ATCGAGGGAAGG) was introduced after the *Bam*H1 site. The rNATx gene was assembled *in vitro* by "oligonucleotide overlapping extension" using the Polymerase Chain Reaction (PCR). For the assembly of the entire gene, the four overlapping oligonucleotides were mixed in a single reaction using the following concentrations: rNATx-Up1 (0.4 pmol/µL), rNATx-Lw4 (0.4 pmol/µL), rNATx-Up2 (0.1 pmol/µL), rNATx Lw-3 (0.1 pmol/µL), and Vent-Pol (NEB) was added. The PCR reaction was conducted at 61 °C during 30 cycles; the product of this reaction was run on a 0.1% agarose gel stained with GelRed dye (Biotium) and detected using ultraviolet (UV) light. After the previous step, the amplified DNA was purified from the agarose gel using the PCR Product Purification (Roche).

For the SATx, four overlapping oligonucleotides were designed to assemble the rSATx gene. Furthermore, two recognition sites for *Bam*H1 and *Pst*I were also added to oligonucleotides forward (rSATx-Up1) and reverse (rSATx-Lw4), respectively. An extra sequence for Factor Xa (ATCGAGGGAAGG) was included after adding the *Bam*Hi site to obtain a mature toxin with no fusion tag (rSATx-Up1). The gene for rSATx toxin was assembled *in vitro* by "oligonucleotide overlapping extension" using the Polymerase Chain Reaction (PCR). To complete the entire sequence of rSATx, four oligonucleotides were mixed and added in the same reaction following the next concentrations: rSATx-Up1 (0.4 pmol/µL), rSATx-Lw4 (0.4 pmol/µL), rSATx-Lw2 (0.1 pmol/µL), rSATx-Up3 (0.1 pmol/µL), and Vent-Pol. The amplification by PCR was conducted at 60 °C for 30 cycles. The final product obtained by PCR was run in an agarose 0.1% gel stained with GelRed® dye (Biotium) and revealed under ultraviolet (UV) light.

The assembled genes (rNATx and rSATx) previously amplified were digested with *Bam*HI and *Pst*I enzymes. They were run and extracted from agarose gels (1.2%) and were ligated to the pQE30 vector, which had been previously cut

with the same enzymes used. Each recombinant vector was independently transformed in *E. coli* XL1-Blue cells using heat shock, and cells were plated in LB agar containing 100 μg/mL ampicillin. Several colonies were picked up for DNA sequencing (pQE-Fwd (GAGCGGATAACAATTATAA) and pQE-Rev (GGTCATTACTGGATCTAT)). For those colonies for which DNA amplification was expected, they were cultured in LB medium containing ampicillin, and their plasmids were purified and again sequenced.

## 2.7 Expression and purification of native and recombinant proteins

Once the genes were linked to pQE30, they were expressed. The transformed cells using each vector (pQE30:rNATx and pQE30:rSATx), the cells that were able to produce the rNATx and rSATx, were grown in Luria-Bertani LB media. The cells were incubated at 37 °C until they reached 0.6 absorbance units (AU) measured at 600 nM. Once the bacterial cultures reached the desired UAs, the media were supplemented with 1 mM isopropyl-β-D-thiogalactopyranoside (IPTG) for 18 h at 16 °C. After the expression time was completed, the culture media were centrifuged (30 min, 9,800 *g*, in a JA-14 rotor); the pellets were reconstituted with washing buffer (0.05 M Tris-HCl, pH 8.0), and the cells were broken using a One-Shot Cell disruptor (Constant Systems, Northants, United Kingdom). The products obtained after disrupting the cells were centrifuged again (22,000 *g* for 20 min). The soluble fraction was discarded, and the insoluble fraction with IBs was kept.

The insoluble fractions were rinsed with washing buffer, and then, they were centrifuged again for 20 min at 18,000 *g* in a JA-20 rotor. The supernatant obtained was discarded and the IB present in the insoluble fraction was dissolved with 0.05 M Tris-HCl (pH 8.0) plus 6 M GndHCl. The products previously obtained were centrifuged at 18,000 *g* in a JA-20 rotor to discard the insoluble materials. The supernatant where the recombinant toxins were concentrated were used to carry out an affinity column chromatography Ni-NTA (Ni-nitrolotriasic acid) which was performed according to manufacturer's instructions (Qiagen, CA, USA), the buffers used: buffer A (6 M GndHCl in a 0.05 M Tris-HCl pH 8.0 buffer) and buffer B (6 M GndHCl in 0.5 M Tris-HCl pH 8.0 plus 400 mM imidazole). A second step of protein purification consisted of Reversed-Phase High Performance Liquid Chromatography (RP-HPLC) using an analytical C$_{18}$ reversed-phase column (Vydac 214 TP 4.6 x 250 mm, Hesperia, CA, USA). The gradient used was a linear one that consisted of solvent A (0.1% trifluoroacetic acid, TFA in water) and solvent B (0.1% TFA in acetonitrile). The gradient used was a linear one that consisted of solvent A (0.1% trifluoroacetic acid, TFA, in water) and solvent B (0.1% TFA in acetonitrile). The gradient was from 20 to 60% of solvent B in 65 min at a flow rate of 1 mL/min. The protein elution was monitored at 230 nm (UV detection). Buffers salts, including GndHCl were eliminated, and the eluted recombinant toxins (rNATx and rSATx) were vacuum dried through a Speed Vac Savant. To isolated native toxins as Ts1 from either *Centruroides* or *Tityus* venoms, 2 mg of dried venom was fractioned using the same RP-HPLC protocols.

## 2.8 Electrophoretic analysis and western blotting

Qualitative analysis of recombinant toxins expression rNATx and rSATx was performed using SDS-PAGE under reducing conditions [28]. The gels were stained with Coomassie Brilliant Blue R-250. Protein samples were separated by SDS-PAGE and transferred to a membrane (polyvinyl difluoride) using 400 mA for an hour in a semi-dry system. After the blotting process, the membrane was blocked with TBST (150 mM NaCl, 10 mM Tris, 0.5% Tween 20, pH 8.0) supplemented with non-fat milk (5%) for 2 h at room temperature. Once the blocking was completed, the buffer previously used was removed, and the membrane was washed three times with TBST. A second step of incubation was required, where the membrane was submerged in anti-histidine monoclonal antibody coupled to horseradish alkaline peroxidase (1:2000) for 1 h at room temperature. After finishing, the membrane was washed three times with TBST to remove the free antibodies. The membrane was revealed by BCIP/NBT (Invitrogen, Waltham, MA, USA) according to the manufacturer's protocols.

## 2.9 Mass spectrometry analysis

The molecular masses of the recombinant and native toxins were determinized using a mass spectrometry analyzer where 500 pmol of each recombinant toxins were reconstituted in 50% acetonitrile with 1% acetic acid in a final volume

of 5 µL. the samples were injected into a Thermo Scientific LCQ Fleet ion trap mass spectrometer (San Jose, CA, USA) using a Surveyor MS syringe pump delivery system. The eluate flow rate of 10 µL/min was reduced, with only 5% of the total sample (0.5 µL/min) directed into the nanospray source. The spray voltage was adjusted to 1.5 kV, and the capillary temperature was kept at 150°C. The fragmentation source was operated with a collision energy of around 25–35 V, a normalized collision energy of 35–45% (arbitrary units), and wideband scanning was activated. All spectra were acquired in positive-ion mode. Data acquisition and deconvolution were carried out using Xcalibur on a Windows NT PC.

## 2.10 Circular dichroism

The secondary structure of each recombinant proteins rSATx and rNATx was carried out by Chirascan spectropolarimeter (Applied Photophysics, Leatherhead, UK). All data obtained from the recombinant proteins were recorded within a wavelength ranging from 190 to 250 nm at 20°C in 0.1 cm-path quartz cells. Data were taken and registered every 1 mm at 50 nm/min. Each protein was prepared in a final concentration of 0.6 mg/mL. The CD values obtained correspond to the mean of three recordings. All CD spectra were deconvoluted employing the Beta Structure Selection webserver (BeStSel).

## 2.11 Biological activities

The neurotoxic activity of rNATx and rSATx was assayed by intracranial administration in mice (18–20 g) following Pedigo *et al.* [29]. Mice were divided in two groups (water as control and toxin) with three mice in each group and 3 µg/ mouse were administrated through the skull at a final volume of 5 µL within a puncture point 2 mm lateral to bregma using a syringe with a truncated 27-gauge needle in a fixed tube, it penetrated the brain only 3 mm from the top of the skull. The two groups were monitored for 48 h after control or sample administration.

## 2.12 Animal immunization

To reach an immune response, groups of rabbits (n = 2) were hyperimmunized subcutaneously with around 10 mg of rSATx and rNATx. The hyperimmunization started administering 0.02 mg up to reach 0.64 mg weekly for each recombinant toxin mixed either with aluminum hydroxide or Incomplete Freund's Adjuvant. The immunization period varied between the toxin groups due to the variation in the immune response generated by them. That is, six months for rSATx and four months for rNATx. After the immunization schedule finished, the serum of rabbits was collected, and the antibodies were purified from plasma through acid precipitation using 5% caprylic acid according to standard procedures [30]. The antibodies were freeze-dried (50 mg/mL), and they were kept at -20 °C and labeled as ready-to-use.

## 2.13 Enzyme-linked immunosorbent assay

Antibodies raised against both recombinant toxins were tested through enzyme-linked immunosorbent assay (ELISA). This assay aimed to determine if the antibodies previously produced could bind to lethal toxins presented in crude venoms of *Centruroides* and *Tityus* scorpions. Promptly, flat-bottom 96 MicroWell poly-styrene microtiter plates (Maxisorp Nunc, Merck KGaA, Darmstadt, Germany) were filled with 100 µL/well at 5 µg/mL using recombinant toxins or crude venom in carbonate/bicarbonate stock solution at pH 9.5 and incubated for 2 h at 37°C. After the incubation step, wells were emptied and washed three times with 200 µL/well of washing buffer containing 50 mM Tris-HCl, 150 mM NaCl, 0.05% Tween 20, and pH 8. After this, wells were supplemented with 150 µL/well of blocking buffer (50 mM Tris-HCl, 5 mg/mL gelatin, 0.2% Tween 20, and pH 8) and incubated for 2 h at 37°C. After the blocking cycle, wells were washed again with the same washing solution previously used. Antibodies anti-rNATx or anti-rSATx were mixed with saline buffer (50 mM Tris-HCl, 0.5 M NaCl, 1 mg/mL gelatin, 0.05% Tween 20, and pH 8). Different amounts of antibodies were used depending on the assay; for instance, 5 µg of anti-rNATx was used to recognize rNATx and whole venoms of *Centruroides* scorpions. In the case of anti-rSATx, 50 and 250 µg was required to recognize rSATx and whole venoms of *Tityus* scorpions, respectively. Antibodies were placed in the first well (150 µL) and serially diluted 1:3 from wells 2–11; well 12 corresponded to a blank

well with saline buffer. Plates were incubated with each antibody, incubated for 1 h at 37°C and washed three times with washing buffer. A peroxidase-conjugated goat anti-rabbit ($5 \times 10^{-4}$ µg/mL, Zymed, Thermo Fisher Scientific Inc., Waltham, MA) was used as a secondary antibody capable of recognizing the first antibody. Then 100 µL/well of this antibody was used and incubated for 1 h at 37°C and washed three times with washing buffer. Finally, 100 µL/well of peroxidase was supplied with a soluble solution BM Blue POD substrate (Merk KGaA, Darmstadt, Germany) and incubated for 10 min at room temperature. Once the time ended, the reaction was blocked with 100 µL/well SDS 5%. The absorbance of each well was measured at 450 nm and analyzed using dose-response analysis in Prism 6.0 (GraphPad, Inc., San Diego, CA) [31].

### 2.14 *In vivo* neutralization test

Mice CD-1 strain weighing around 18–20 g were injected intravenously (IV) using the tail vein. The samples used in this assay consisted of mixing IgGs obtained from rabbits with venom or native toxins. The experiments followed the guidelines of Animal Welfare, using the minimum number of animals. For neutralization tests, $3LD_{50}$ of whole venoms of *C. limpidus, C. tecomanus, C. suffusus,* and *C. noxius* were incubated with anti-rNATx, whereas *T. jaimei, T. festae,* and *T. serrulatus* were incubated with anti-rSATx. Venoms and antibodies were incubated for 30 min at 37 °C, and they were administered by the IV route. The animals, after administration, were observed for 48 h.

### 2.15 Peripheral blood mononuclear cells (PBMCs) isolation

Human healthy PBMCs were isolated from whole blood by using the Ficoll-Paque method reported by the reagent provider (GE-Healthcare Bio-Sciences AB, Uppsala, Sweden). Fresh PBMCs were maintained in complete RPMI-1640 media (cRPMI), supplemented with penicillin/streptomycin antibiotics (GIBCO, Life Technologies, Co.) and 10% AB serum**..** The racial/ethnic distribution of the donors was as follows: Female Latine (36 years), Female Latine (31 years), and Female Latine (30 years). Donors ranged in age from 30 to 36 years, with a mean age of 32.3 years.

### 2.16 PBMCs stimulation with scorpion toxins

The experiment was standardized for each treatment with 500,000 cells (PBMCs) in 200 µL volume, on cRPMI media for 24 h incubation, at 37°C in a CO2 atmosphere. Scorpion toxins were resuspended in supplemented media RPMI-1640. All of them were assayed at 1, 10, and 20 µg/200 µL reaction volume. As negative controls, PBMCs were maintained in cRMPI media without stimuli. All experiments were done in triplicate using PBMCs obtained from healthy donors.

### 2.17 T-cell immunophenotype and absolute cell count

T-cell subsets were determined using BD Multitest CD3/CD4/CD8/CD45 (Ref 340499, Becton Dickinson, NJ), and absolute cell count was carried out using BD Counting beads (Ref 355925, Becton Dickinson, NJ). Sample preparation and analysis were performed following the manufacturer's instructions. Samples were acquired in the BD FACSCanto II cytometer and analyzed in FlowJo (FlowJo version 10.10; BD Science, San Diego, CA, USA) and Kaluza software (version 2.4, Beckman Coulter Life Sciences, CA) where the gate strategy to identify the populations was determined.

### 2.18 Statistics

The Prism 6.0 software (GraphPad Inc., San Diego, CA) was used to analyze the data. For the ELISA, nonlinear regression analyses with a variable slope were performed. Prism 6.0 software was also employed for all statistical analyses, including the calculation of mean values, standard deviations, coefficients of variation, and 95% confidence intervals. Also, $EC_{50}$ values were estimated from individual curve fits and compared between antibodies under identical experimental conditions using an extra sum-of-squares F-test, with differences considered statistically significant when $p < 0.05$.

# 3 Results

## 3.1 Protein alignments for designing the consensus toxins SATx and NATx

Several proteins coding for neurotoxins were found in the NCBI and Uniprot databases. Those primary structures from the *Centruroides* and *Tityus* genera were selected and curated based on their toxicity data. Most of the toxic structures were selected, aligned using the T-Coffee online server, and the Jalview program **Table 1**.

## 3.2 Gene construction

The synthetic genes for rNATx and rSATx were constructed and amplified. The N-terminus had an extra sixteen amino acids (MRGSHHHHHHGSIEGR) that include a histidine label (6xHis), a restriction site for *Bam*HI, and a proteolytic cleavage site. Consequently, the recombinant proteins, rNATx and rSATx, ended with 82 and 78 residues, respectively.

## 3.3 Screening of rNATx and rSATx toxin expression

Each gene rNATx or rSATx was cloned into the expression vector pQE30 and transformed into *E. coli* Origami strains. After protein expression, SDS-PAGEs were performed, and protein bands around 10 kDa were observed, corresponding to the rNATx and rSATx. Western blot analyses were carried out using the same protein samples from the SDS-PAGE

**Table 1. Protein alignment of β-toxins found in *Centruroides* and *Tityus* scorpions and consensus sequences.**

**A)**

| Toxin | Amino acid toxin sequence[1] | Uniprot code | References |
|---|---|---|---|
| Td1 | -KDGYLMEPNG**C**KRG**C**LTRPARY**C**PNE**C**SRLKGKDGY**C**YLWLA**C**Y**C**YNMPESAPVWERATNR**C**G-K | QIL180 | [32] |
| Ts1 | -KEGYLMDHEG**C**KLS**C**FIRPSGY**C**GRE**C**GIKKGSSGY**C**A-WPA**C**Y**C**YGLPNWVKVWDRATNK**C**-- | P15226 | [33] |
| Tppa1 | -KDGYLVGNDG**C**KYS**C**LTRPGHY**C**ASE**C**SRVKGKDGY**C**YAWMA**C**Y**C**YSMPDWVKTWSRSTNR**C**G-R | C0HLZ0 | [34] |
| Tppa2 | -KDGYLVGNDG**C**KYS**C**LTRPGHY**C**ASE**C**SRVKGKDGY**C**YAWMA**C**Y**C**YNMPNWVKTWSRATNK**C**--- | C0HL21 | |
| Tce3 | -KDGYIIEHRG**C**KYS**C**FFGSSSW**C**NKE**C**TLKKGSSGY**C**A-WP**C**W**C**YGLPDSVKIFDSNNNK**C**SKK | C0HLZ2 | |
| Tt1g | -KEGYLMDHEG**C**KLS**C**FIRPSGY**C**GRE**C**AIKKGSSGY**C**A-WPA**C**Y**C**YGLPNWVKVWERATNR**C**--- | P0DMM8 | [35] |
| Tst1 | GKEGYLMDHEG**C**KLS**C**FIRPSGY**C**GRE**C**TLKKGSSGY**C**A-WPA**C**Y**C**YGLPNWVKVWDRATNK**C**--- | P56612 | [36] |
| Tb1 | -KEGYLMDHEG**C**KLS**C**FIRPSGY**C**GSE**C**KIKKGSSGY**C**A-WPA**C**Y**C**YGLPNWVKVWDRATNK**C**--- | P56611 | |
| SATx | -KEGYLMDHEG**C**KLS**C**FIRPSGY**C**GSE**C**SRKKGSSGY**C**AAWPA**C**Y**C**YGLPNWVKVWDRATNK**C**--- | – | This work |

**B)**

| Toxin | Amino acid toxin sequence[1] | Uniprot code | References |
|---|---|---|---|
| Cse1 | KDGYLVEK-TG**C**KKT**C**YKLGENDF**C**NRE**C**KWKHIGGSYGY**C**YGFG**C**Y**C**EGLPDSTQTWPLPNKT**C**- | P01491 | [37] |
| Cl13 | KEGYLVDYHTG**C**KYT**C**AKLGDNDY**C**VRE**C**RLRYYQSAHGY**C**YAFA**C**W**C**THLYEQAVVWPLPNKR**C**K | C0HK69 | [38] |
| Cll1 | KEGYLVNKSTG**C**KYG**C**FWLGKNEN**C**DKE**C**KAKNQGGSYGY**C**YSFA**C**W**C**EGLPESTPTYPLPNKS**C**S | P45667 | [39] |
| Cll2 | KEGYLVNHSTG**C**KYE**C**FKLGDNDY**C**LRE**C**KQQYGKGAGGY**C**YAFG**C**W**C**NHLYEQAVVWPLPKKT**C**N | P59898 | |
| Cell9 | KEGYLVNHSTG**C**KYE**C**FKLGDNDY**C**LRE**C**RQGYGKGAGGY**C**YAFG**C**W**C**THLYEQAVVWPLPKKT**C**N | P0CH41 | [40] |
| Ct1a | KEGYLVNHSTG**C**KYE**C**FKLGDNDY**C**LRE**C**RQQYGKGAGGY**C**YAFG**C**W**C**THLYEQAVVWPLPKKT**C**N | P18926 | [41] |
| Cn2 | KEGYLVDKNTG**C**KYE**C**LKLGDNDY**C**LRE**C**KQQYGKGAGGY**C**YAFA**C**W**C**THLYEQAIVWPLPNKR**C**S | P01495 | [42] |
| Csem1 | KEGYLVNSYTG**C**KYE**C**LKLGDNDY**C**LRE**C**RQQYGKS-GGY**C**YAFA**C**W**C**THLYEQAVVWPLPNKT**C**N | P56646 | [43] |
| Css2 | KEGYLVSKSTG**C**KYE**C**LKLGDNDY**C**LRE**C**KQQYGKSSGGY**C**YAFA**C**W**C**THLYEQAVVWPLPNKT**C**N | P08900 | [44] |
| NATx | KEG**Y**LVNHSTG**C**KYE**C**LKLGDNDY**C**LRE**C**KQQYGK**G**AGGY**C**YAFG**C**W**C**THLYEQAVVWPLPNKT**C**N | – | This work |

[1] Toxins were obtained from the database Uniprot. Amino acid sequences of the most lethal mammalian scorpion toxins were selected. A) toxins correspond to the *Tityus* genus, and B) toxins correspond to the *Centruroides* genus. Both consensus sequences (NATx and SATx) were obtained after protein alignment. Immunogenic motifs in SATx and NATx are underlined; those regions were known using Kolaskar and Tongaonkar (1990).

gels. The recombinant proteins (~10 kDa) were visualized through commercial antibodies able to recognize the His-tag (**Fig 1**).

### 3.4 Expression and purification of heterologous toxins

The colonies previously tested for expressing each toxin were stored in 15% of glycerol and kept at -80 °C and labeled as ready-to-use. These cells were grown in LB medium, inducing the protein expression with 1 mM IPTG. After disrupting cells, both toxins were concentrated in IB, which was dissolved using a chaotropic agent, and then, the dissolved IB was passed through an affinity column (**Fig 2**).

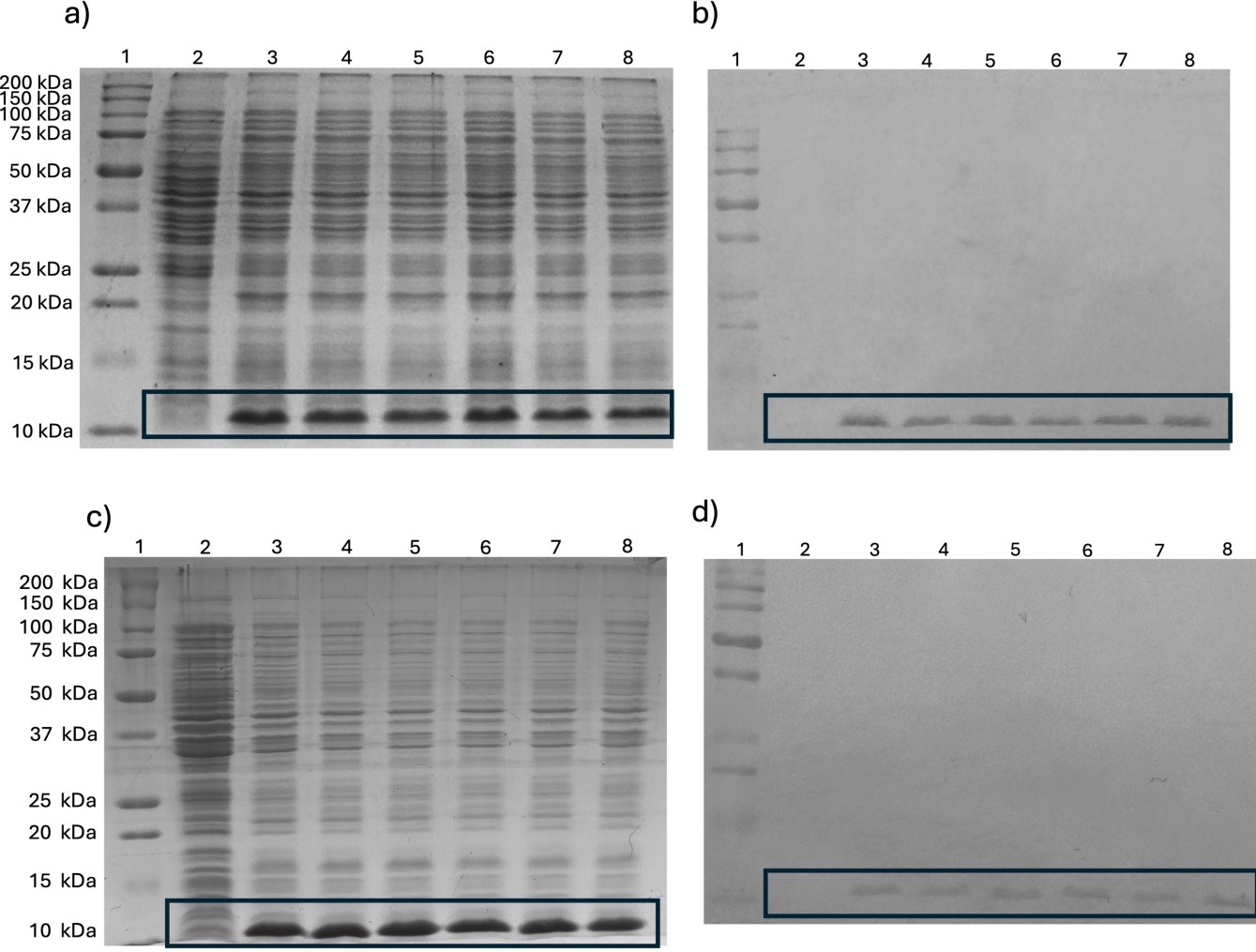

**Fig 1. Electrophoretic profile of rNATx and rSATx, including a western blot analysis. a)** An SDS-PAGE 15% was carried out, where line 1 represents the molecular weight markers, 2. *E. coli* without inducer (IPTG), and the lines from 3 to 8 show several *E. coli* colonies that were induced with IPTG 1 mM, producing the rNATx around 10 kDa. **b)** A western blot was performed using the same samples previously used; the transfer was incubated with a monoclonal anti-histidine antibody, and bands around 10 kDa were visible using the antibodies. **c)** An SDS-PAGE 15% was performed, where line 1 corresponds to the molecular weight marker, 2. *E. coli* cells without inducer and lines 3 to 8 belong to those cells that were supplemented with inducer (1 mM IPTG), expressing the rSATx visualized at 10 kDa. **d)** A western blot was performed using the previous samples and the transfer was incubated with a monoclonal antibody for His-tag recognition, and bands around 10 kDa showed up, belonging to the expression of rSATx. Boxes indicate the expression and detection of recombinant proteins.

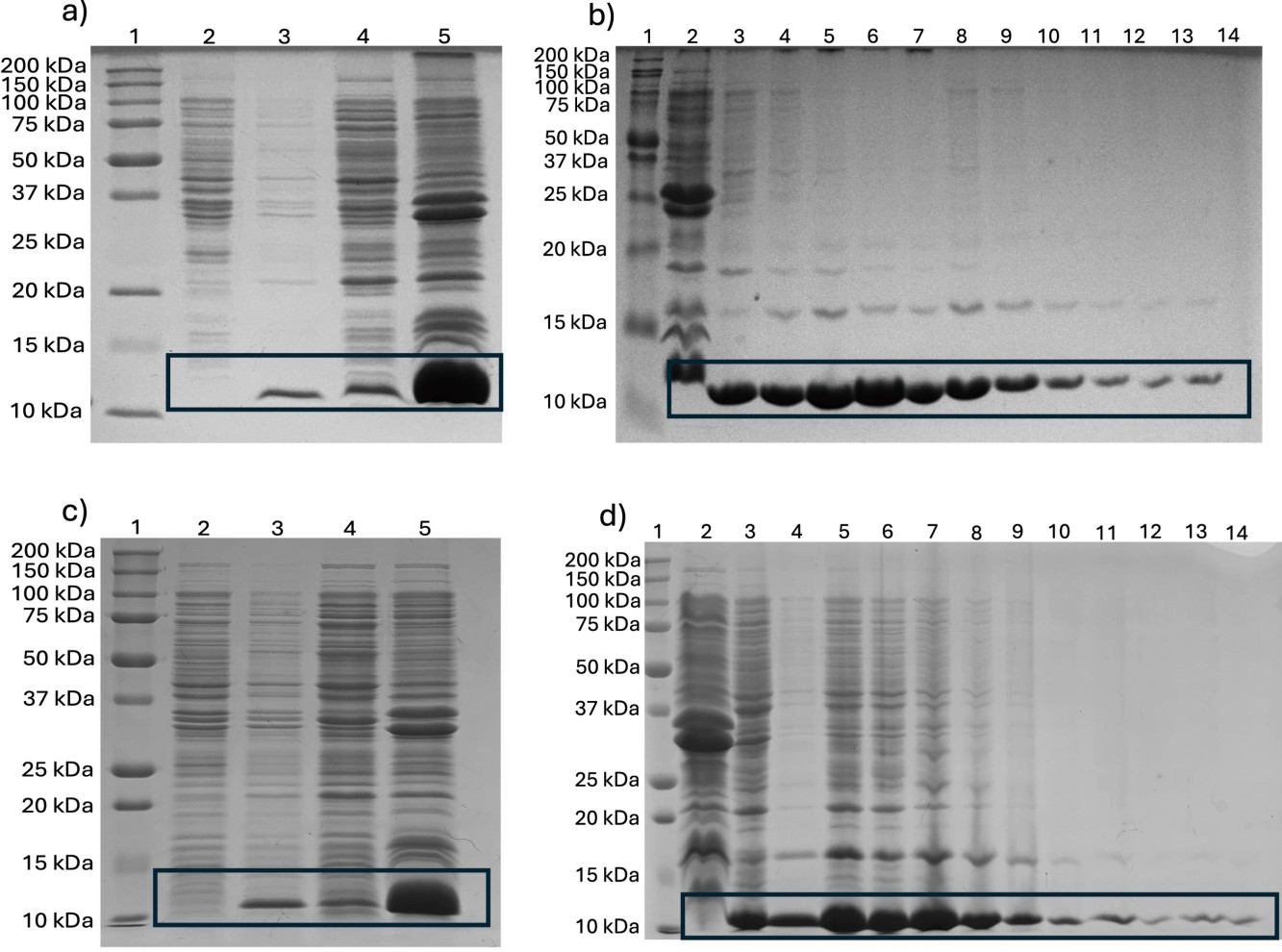

**Fig 2. Purification of inclusion bodies by affinity chromatography.** After disrupting the cells, both toxins were accumulated in inclusion bodies (IB), a) rNATx, a 15% SDS-PAGE, where line 1 is the molecular weight markers, line 2 and 3 are *E. coli* without and with IPTG, respectively, line 4 corresponds to the soluble fraction, and line 5 is rNATx IB. **b)** Affinity purification of rNATx IB, line 1, molecular weight marker, line 2, recirculating proteins, line 3, first wash with 30 mM imidazole, lines 4-14, protein elution using 400 mM imidazole. c) rSATx, a 15% SDS-PAGE, where line 1 is the molecular weight markers, line 2 and 3 are *E. coli* without and with IPTG, respectively, line 4 corresponds to the soluble fraction, and line 5 is rSATx IB. **d)** Affinity purification of rSATx IB, line 1, molecular weight marker, line 2, recirculating proteins, line 3, first wash with 30 mM imidazole, lines 4-14, protein elution using 400 mM imidazole. Boxes indicate the expression of recombinant proteins.

After having rNATx and rSATx purified by affinity chromatography, the samples were directly purified using two steps of Reversed-Phase High Performance Liquid Chromatography (RP-HPLC). Isoforms of both toxins, rNATx and rSATx, were found in the first step of purification. After that, a second step of purification was required to obtain the active fraction (**Fig 3**). Both expressed toxins were subjected to mass spectrometry analysis, having experimental molecular masses of 9,370.0 and 8,812.0 Da for rNATx and rSATx, respectively. Those molecular masses represent an oxidized form of them. Isoforms expressed were reduced using DTT, and the products after the reduction process were analyzed by mass spectrometry, showing 9,376.6 and 8,820.9 Da corresponding to rNATx and rSATx, respectively, in their reduced forms. The recombinant protein yields after protein purification steps, for rNATx and rSATx, were 1 and 0.5 mg/L, respectively.

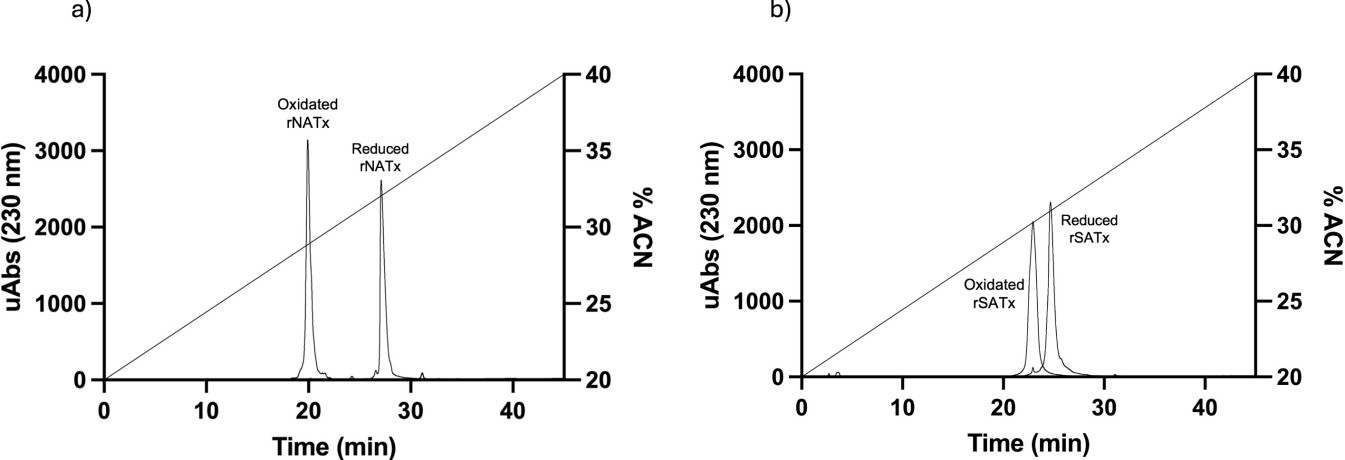

**Fig 3. RP-HPLC of rNATx and rSATx. a)** active isoform and reduced state of rNATx. **b)** active isoform and reduced state of rSATx. The purification steps were applied using an analytical C$_{18}$ reversed-phase column (Vydac 214 TP 4.6 x 250 mm, Hesperia, CA, USA) through a linear gradient using solvent A (0.1% trifluoroacetic acid, TFA, in water) and solvent B (0.1% TFA in acetonitrile). The linear gradient started from 20% solvent B for 45 min at a flow rate of 1 mL/min and was monitored with UV detection at 230 nm.

## 3.5 Biological activity of recombinant toxins

Active isoforms were tested in the CD-1 strain; the samples were administered intracranially (**Table 2**). Isoforms of rNATx and rSATx could produce acute envenoming. After evaluating their toxicity activity, those active toxins were chosen to carry out the antibody production.

## 3.6 Secondary structure of rNATx and rSATx

Secondary structure deconvolution of the CD spectra indicates that the three recombinant toxins share a broadly comparable fold at the level of secondary structure (**Fig 4**). The estimated α-helix content decreases slightly from 12.4, 10.0, and 8.9% for rNATx, rSATx, and rCssII, respectively. In contrast, the β-sheet content spans a wider range, with values of 27.8, 34.5, and 27.4% for rNATx, rSATx, and rCssII, respectively. The fraction assigned to disordered regions remains high and relatively similar in all three samples, that is, 44.7, 49.3, and 48.7% for rNATx, rSATx, and rCssII, respectively. Here, rCssII was included as a positive control since its in vitro folding and active conformation have been previously reported [46]. The CD spectrum of rNATx is similar and consistent with that of rCssII, whose structure was included to design NATx. Accordingly, both their spectral similarities and their closely matched β-sheet content of rNATx and rCssII support the notion that rNATx is well folded. On the other hand, rSATx exhibits the most pronounced differences in its secondary

**Table 2. Toxic activity of recombinant toxins in mice.**

| Toxin | Amount tested (µg) | Alive/ Dead | Survival rate |
|---|---|---|---|
| rNATx (oxidated) | 3 | 0/3 | 0% |
| rNATx (reduced) | 3 | 3/0 | 100% |
| PBS | – | 3/0 | 100% |
| rSATx (oxidated) | 3 | 0/3 | 0% |
| rSATx (reduced) | 3 | 3/0 | 100% |
| PBS | – | 3/0 | 100% |

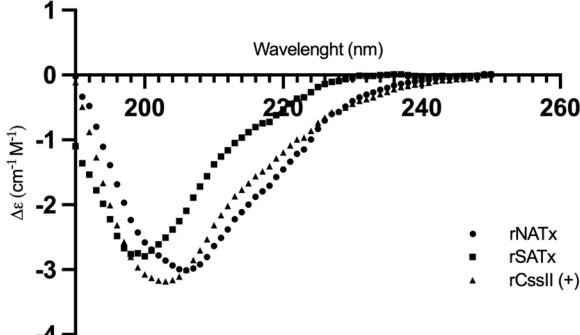

**Fig 4. Circular dichroism of recombinant proteins rNATx, rSATx, and rCssII.** The secondary structure analysis of active isoforms of rNATx, rSATx, and rCssII of the CD spectra was analyzed according to Micsonai [45].

structure and shows the least spectral overlap compared to rNATx and rCssII. The results nevertheless indicate that it is also well folded.

### 3.7 Rabbit immunization and antibody recognition

Groups of rabbits were immunized with either rNATx or rSATx supplemented with adjuvants. The administration of the recombinant toxins began with 0.02 mg of each peptide, and the amount was increased by 0.02 mg every week, up to a total of 0.64 mg. After completing the immunization process, rabbits were bled, and the entire serum was removed from the total blood by centrifugation. Immunoglobulins were then separated from other non-desirable proteins, such as albumin, by caprylic precipitation. The remnant of caprylic acid was removed through a dialysis step, and the immunoglobulins were freeze-dried and stored at a concentration of 50 mg/mL. The antibodies' ability to bind to native toxins presented in the whole venom of *Centruroides* or *Tityus* scorpions was evaluated in an ELISA assay; these results are shown below (**Fig 5**).

Anti-rNATx ($F_{(16, 130)} = 4.146$, $p < 0.0001$) and anti-rSATx ($F_{(12, 104)} = 4.151$, $p < 0.0001$) recognize *Centruroides* and *Tityus* venom toxins, respectively. Anti-rNATx had a better response and consistent recognition than anti-rSATx. Under the same experimental conditions, anti-rNATx antibodies required lower concentrations to reach half of the maximum response ($EC_{50}$ value) compared to anti-rSATx (**Fig 6**).

These data correlate with the *in vivo* trials, where anti-rNATx showed better venom neutralization (**Table 3**).

### 3.8 Venom and toxin neutralization

The efficacy and quality of antivenoms are based on their ability to neutralize the lethal toxins found in whole venoms. Consequently, anti-rNATx and rSATx were challenged for their ability to neutralize native toxins and whole venoms. The antibodies produced in this work were incubated and tested with 3 $LD_{50}$, anti-rNATx were challenged with venoms from *C. limidus, C. noxius, C. tecomanus, C. suffusus,* and with the CssII native toxin isolated from *C. suffusus,* whereas anti-rSATx was challenged with the venoms from *T. jaimei, T. festae, T. serrulatus,* and native Ts1 toxin obtained from *T. serrulatus* (**Table 3**). Antibodies against *Tityus* scorpions neutralized partially most venoms used, being the *T. festae* the only one neutralized entirely. The native Ts1 was successfully neutralized for this antibody. The anti-rNATx showed better neutralization, as it neutralized the 3$LD_{50}$ of venoms from *C. limidus, C. noxius, C. tecomanus, C. suffusus,* and the native CssII. To reach the total neutralization of *Tityus* venoms, up to 20 mg of anti-rSATx was required, whereas to obtain 100% neutralization of *Centruroides* venoms, up to 1.5 mg of anti-rNATx was required. As part of a neutralization test, anti-rNATx and anti-rSATx were unable to neutralize the venoms of *Tityus* and *Centruroides*. There was no cross-neutralization effect among scorpion genera.

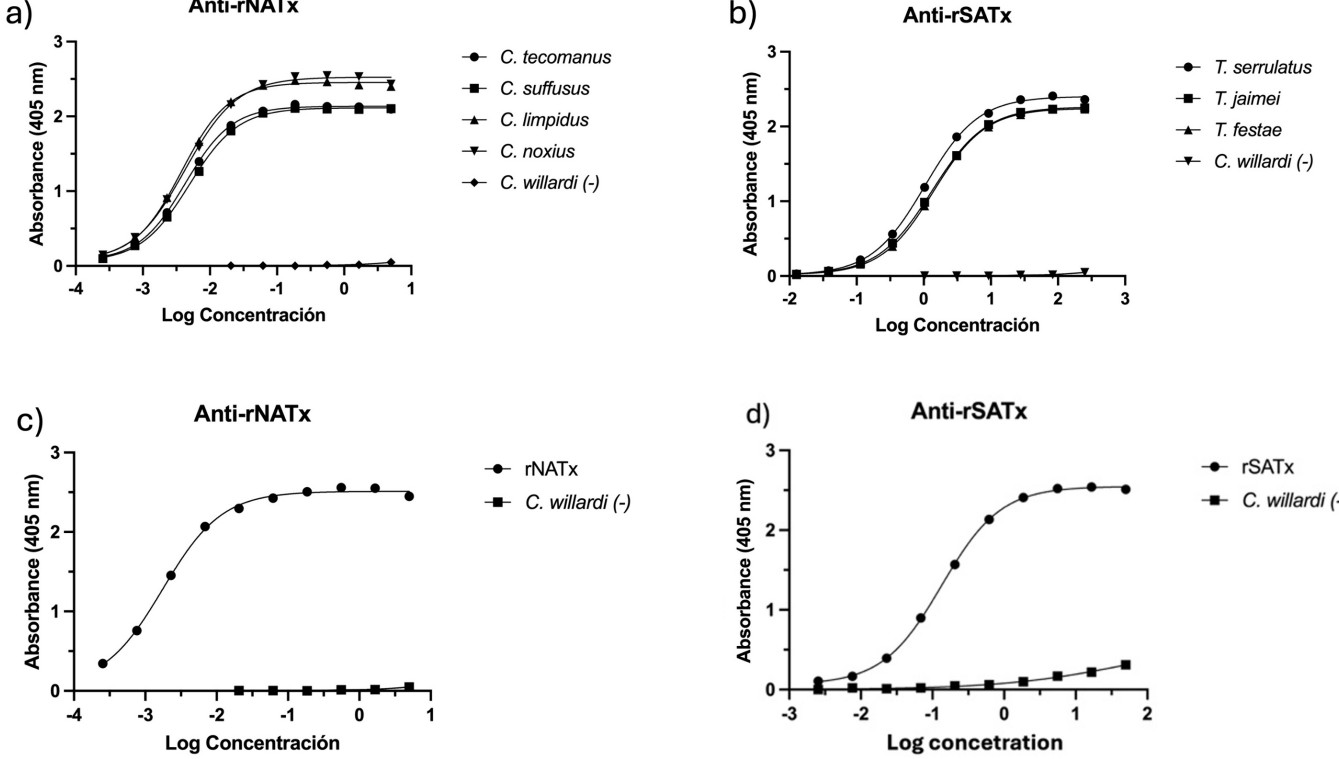

**Fig 5. Antibody recognition of rabbit IgGs to *Centruroides* and *Tityus* species. a)** $EC_{50}$ and CI in parentheses for *C. tecomanus, C. suffusus, C. limpidus, C. noxius,* and rNATx were 0.0042 (0.0039 − 0.0046), 0.0049 (0.0045 − 0.0053), 0.0037 (0.0034 − 0.0040), 0.0042 (0.0039 − 0.0046), 0.0017 (0.0011 − 0.0022), respectively. **b)** $EC_{50}$ and CI in parentheses for *T. serrulatus. T. jaimei, T. festae,* and rSATx were 1.04 (0.94 − 1.27), 1.32 (1.13 − 1.53), 1.39 (1.28 − 1.50), 0.13 (0.12 − 0.14), respectively. c) and **d)** Venom of the Mexican rattlesnake *Crotalus willardi* was used as a negative control in each ELISA experiment. Points represent the mean of triplicates.

### 3.9 *In vitro* immunogenic response of rNATx and rSATx

Human peripheral blood mononuclear cells (PBMCs) isolated from healthy adult donors were exposed to the scorpion toxins rNATx and rSATx under standardized conditions ($5 \times 10^5$ cells per 200 µL) and incubated for 24 h. Following stimulation, T-cell subsets were quantified by flow cytometry using a validated CD3/CD4/CD8/CD45 panel in combination with counting beads, allowing simultaneous assessment of immunophenotype and absolute cell numbers (**Fig 7**).

Stimulation with rSATx at concentrations of 1, 10, and 20 µg per reaction volume did not induce substantial changes in total T-cell numbers or in the absolute counts of CD3⁺, CD4⁺, or CD8⁺ T-cell subsets when compared with unstimulated controls (**Fig 8**). These findings suggest that, under the experimental conditions tested, SATx does not exert a marked effect on T-cell abundance. In contrast, exposure to rNATx resulted in dose-dependent alterations in T-cell populations. While low and intermediate concentrations produced minimal effects, stimulation with 20 µg NATx was associated with a modest but consistent increase in the absolute numbers of total CD3⁺ T cells, as well as CD4⁺ and CD8⁺ subsets. Notably, the most prominent change was observed within the CD4⁺CD8⁺ double-positive (DP) T-cell compartment, which exhibited an increase at 10 µg rNATx, followed by a relative attenuation at the highest toxin concentration.

## 4 Discussion

The incidence of scorpion stings is particularly high in tropical and subtropical areas. In America, the scorpion species that cause most accidents are from the genera *Centruroides* and *Tityus* [52–54]. The use of the whole venom of scorpions

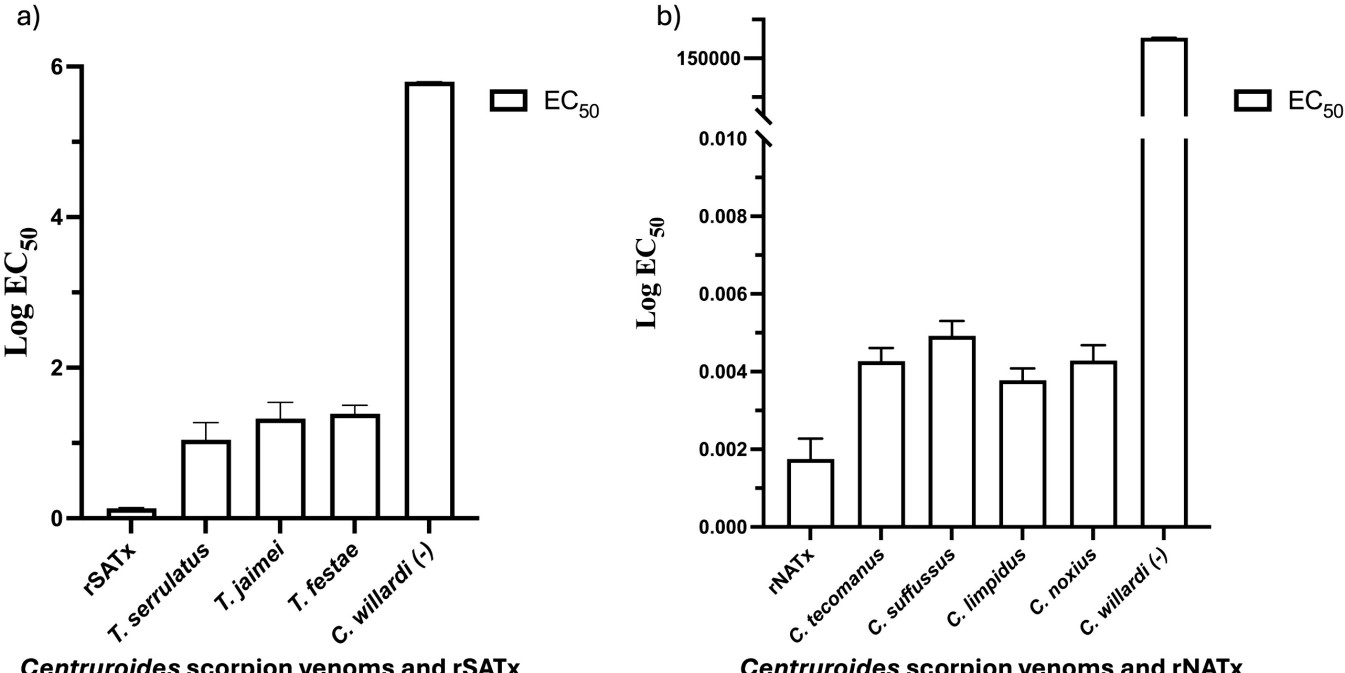

**Fig 6. EC$_{50}$ values obtained from anti-rNATx and anti-rSATx. a)** EC$_{50}$ values using anti-rNATx against *Centruroides* scorpion venoms; **b)** EC$_{50}$ values using anti-rSATx against *Tityus* scorpion venoms. Venom of the Mexican rattlesnake *C. willardi* was used as a negative control. All ELISA assays were made in triplicate, and data were obtained through dose-response analyses.

in antivenom production could limit their efficacy, and one reason is the low concentration of toxins lethal to humans and other mammals [55]. Since the implementation of recombinant scorpion toxins and their use as immunogens for antibody production has been well-documented, here we design two scorpion toxins to observe their antigenicity in rabbits and the performance of the antibodies obtained in neutralizing American buthid scorpion venoms [34,56–59].

Based on previous work, two consensus scorpion toxins were expressed in a heterologous system for immunization protocols. The antibodies were expected to have a broad spectrum of recognition and neutralization of scorpion toxins and whole venoms [24,25,60,61]. The design of rNATx and rSATx was simple and based on aligning toxins with known biological activity from *Centruroides* and *Tityus* species. Both toxins rNATx and rSATx were predominantly expressed as inclusion bodies. Following solubilization with a chaotropic agent, the proteins were purified [23,62]. The active isoforms of both toxins were successfully isolated and used in subsequent immunological assays. The overall yield for each recombinant toxin was approximately 4–5 mg per liter of culture medium. The immunization protocol followed previous experiences with scorpion and elapid toxins used as immunogens in rabbits.

The venoms from *T. jaimei* and *T. serrulatus* were not efficiently neutralized when anti-rSATx was used. On the other hand, all the venoms from *Centruroides* scorpions were successfully neutralized when challenged with the anti-rNATx. During the immunization process, the immune response against rNATx was better following high titers than that against rSATx. This phenomenon can be attributed to some factors, such as the level of immunogenicity of toxins that awakens the host's immune response. Toxins from the genus *Centruroides* seem to be more immunogenic than those from the genus *Tityus*. Similar results were found when comparing titers with venoms of the genus *Buthus*, *Leiurus,* and *Androctonus*, from the Middle East and North Africa [24]. In a previous study conducted in our research group, a set of *Tityus* recombinant scorpion toxins was expressed and used to raise antibodies. After several months of an immunization schedule, the antibody titers did not reach high levels, although those antibodies could neutralize some *Tityus* venoms [34].

**Table 3. Scorpion venom neutralization using rNATx or rSATx.**

| Venom or toxin | LD$_{50}$ µg/mice | Number of LD$_{50}$ tested | Antibody used | Antibody (mg) | Surviving mice/ total mice | µg V/ mg IgG |
|---|---|---|---|---|---|---|
| Neutralization of venoms from *Tityus* species | | | | | | |
| *T. jaimei* | 90[a] | 2 | No antibody (control) | 0 | 0/3 | ------- |
| *T. jaimei* | 90 | 2 | Anti-rSATx | 20 | 1/3 | > 4.0 |
| *T. festae* | 98[b] | 2 | No antibody (control) | 0 | 0/3 | ------- |
| *T. festae* | 98 | 2 | Anti-rSATx | 20 | 3/3 | 9.8 |
| *T. serrulatus* | 39[c] | 2 | No antibody (control) | 0 | 0/3 | ------- |
| *T. serrulatus* | 39 | 2 | Anti-rSATx | 20 | 2/3 | >3.9 |
| Native Ts1 | 5.8[c] | 3 | No antibody (control) | 0 | 0/3 | ------- |
| Native Ts1 | 5.8 | 3 | Anti-rSATx | 5 | 3/3 | 3.5 |
| Native Ts1 | 5.8 | 3 | Anti-rNATx | 5 | 0/3 | ------- |
| Neutralization of venoms from *Centruroides* species | | | | | | |
| *C. limpidus* | 15[d] | 3 | No antibody (control) | 0 | 0/3 | ------- |
| *C. limpidus* | 15 | 3 | Anti-rNATx | 1 | 3/3 | 45.0 |
| *C. noxius* | 5[e] | 3 | No antibody (control) | 0 | 0/3 | ------- |
| *C. noxius* | 5 | 3 | Anti-rNATx | 1.5 | 3/3 | 10 |
| *C. tecomanus* | 10.2[d] | 3 | No antibody (control) | 0 | 0/3 | ------- |
| *C. tecomanus* | 10.2 | 3 | Anti-rNATx | 1 | 3/3 | 30.6 |
| *C. suffusus* | 8.6[d] | 3 | No antibody (control) | 0 | 0/3 | ------- |
| *C. suffusus* | 8.6 | 3 | Anti-rNATx | 1 | 3/3 | 25.8 |
| Native Cn2 | 0.4[f] | 3 | No antibody | 0 | 0/3 | ------- |
| Native Cn2 | 0.4 | 3 | Anti-rNATx | 1 | 3/3 | 1.2 |
| Native Cn2 | 0.4 | 3 | Anti-rSATx | 5 | 0/3 | ------- |

The LD$_{50}$ values correspond to µg of venoms or toxins per 20 g of mouse. [a] Data taken from [47]; [b] Data taken from [48]; [c] Data taken from [22]; [d] Data taken from [49]; [e] Data taken from [50]; [f] Data taken from [51].

An interesting phenomenon has been observed when Na$^+$ channel-targeting neurotoxins from the *Centruroides* genus are used for immunization: high levels of antibody titers are generated throughout the immunization schedule [23,57]. A similar response is documented in this report, where higher antibody titers were produced against rNATx compared to rSATx.

Although the exact mechanisms triggering this heightened immune response are not fully understood, several factors may contribute. Following a scorpion sting, a massive release of neurotransmitters occurs, activating both the sympathetic and parasympathetic nervous systems. Beyond this neurological stimulation, venom components also interact directly with cells of the immune system. These toxin-receptor interactions can activate both pro-inflammatory and anti-inflammatory signaling pathways, contributing to severe symptoms such as pulmonary edema and potentially leading to death [63–68].

Although DP T cells constitute a minor population in peripheral blood, they are increasingly recognized as a functionally heterogeneous subset with the capacity to display helper, cytotoxic, and immunoregulatory properties. These cells have

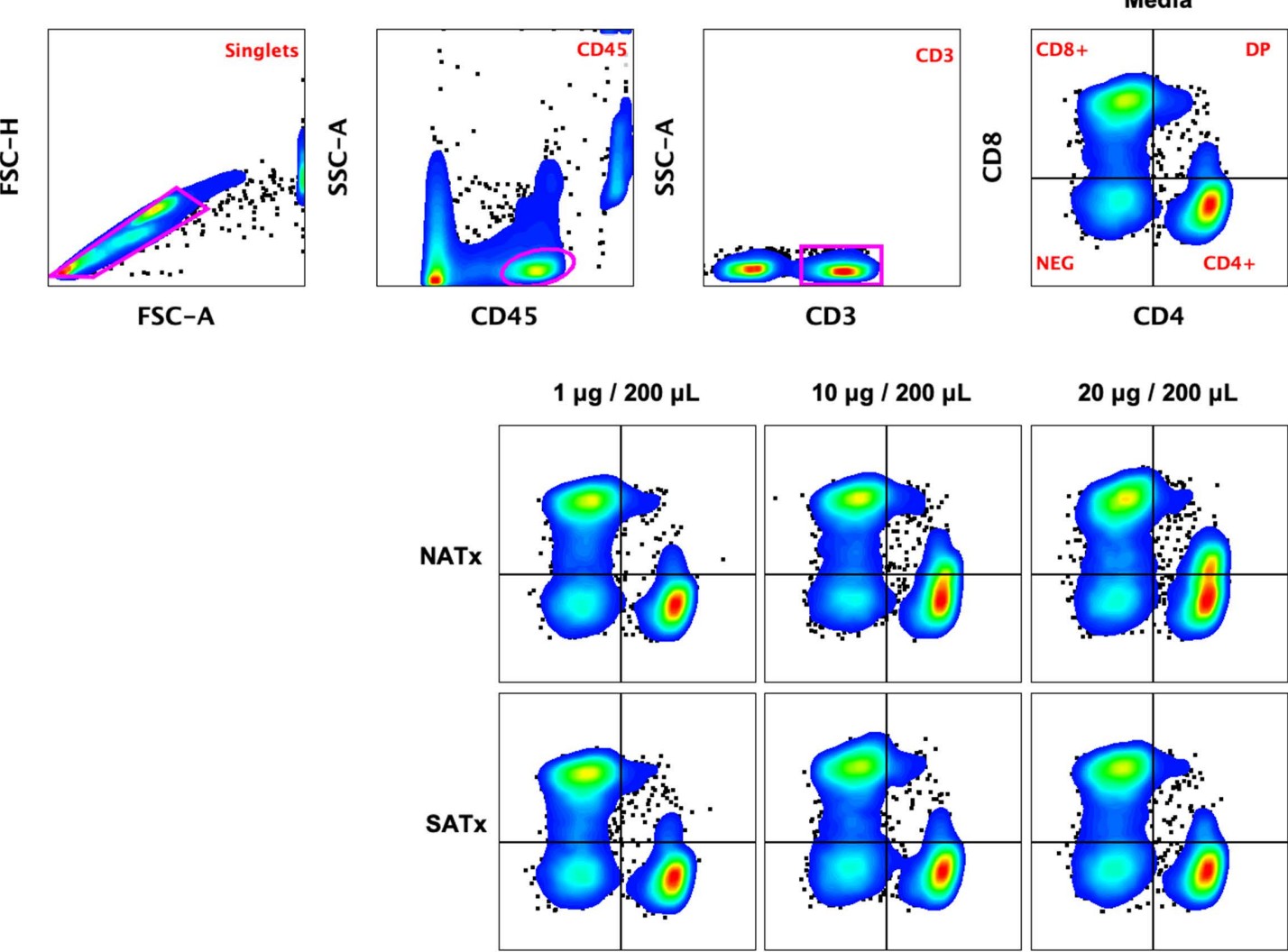

**Fig 7. T-cell immunophenotype analysis. Representative flow-cytometry plots illustrating the T-lymphocyte immunophenotyping strategy.** Events were first gated on singlets based on FSC-H versus FSC-A, followed by identification of total leukocytes using SSC versus CD45 expression. CD3⁺ T-lymphocytes were subsequently gated, and T-cell subsets were resolved based on CD4 and CD8 expression. Quadrant gating was applied to define CD4⁺, CD8⁺, double-positive (CD4⁺CD8⁺), and double-negative populations. Representative density plots are shown for control conditions and experimental treatments.

been reported to expand in contexts of chronic immune stimulation, including persistent infections and malignancies, and are thought to represent antigen-experienced or highly activated memory T cells. The observed modulation of the DP T-cell compartment following rNATx exposure raises the possibility that this toxin may preferentially influence activation or differentiation pathways within this subset, potentially reflecting an early inflammatory or stress-associated immune response. However, additional functional and mechanistic studies will be required to determine whether these changes translate into altered cytokine production, cytotoxic potential, or immunoregulatory activity.

Upon exposure to antigens, the vertebrate immune system responds by producing antibodies through a cascade of cellular interactions. Initially, antigen-presenting cells (APCs) present processed antigens to T cell receptors (TCRs), thereby activating helper T cells (Th cells). These Th cells assist B cells in antibody production by secreting cytokines. Among the

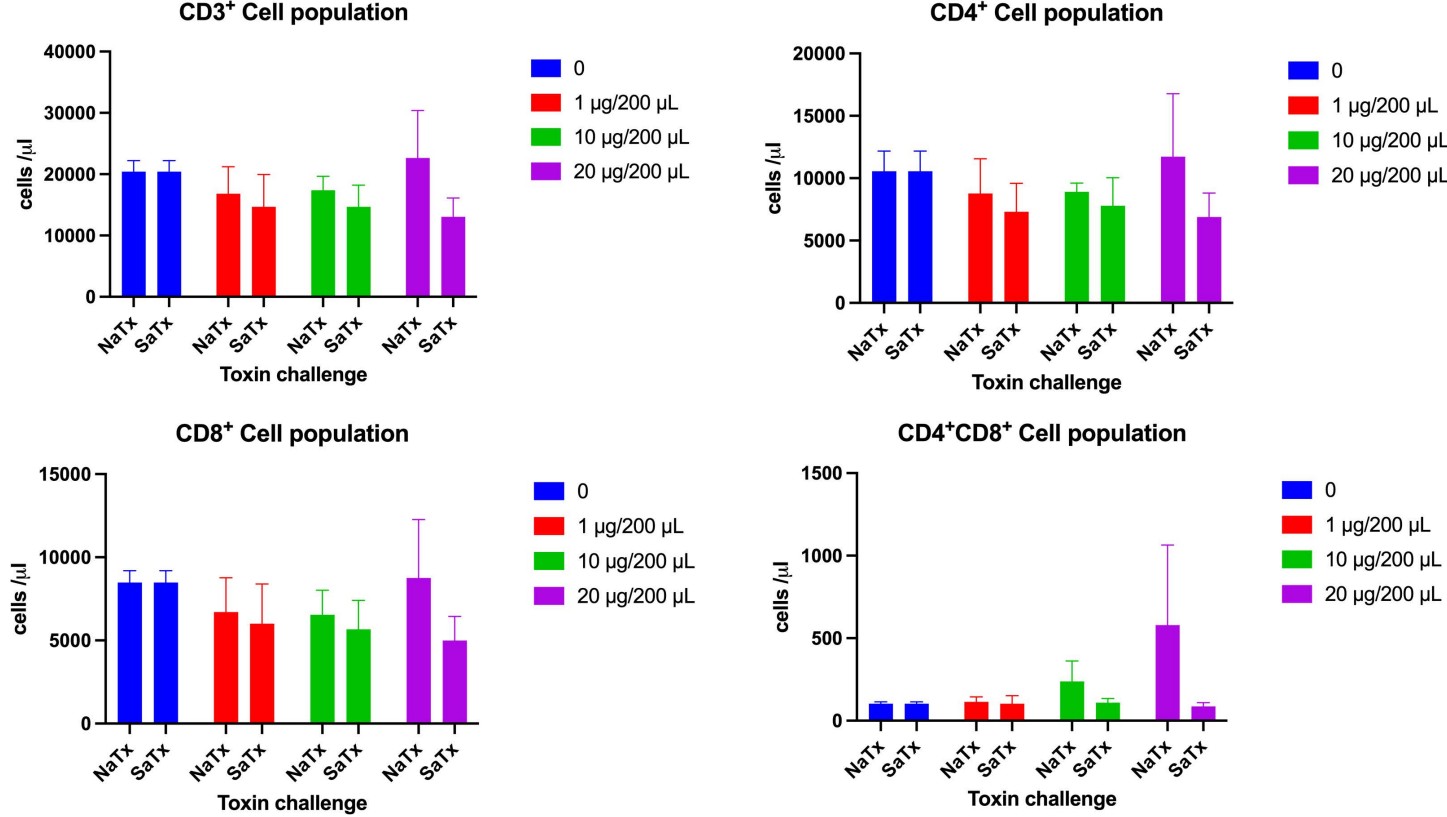

**Fig 8. Effects of rNATx and rSATx on T-cell absolute counts and subset distribution.** Human peripheral blood mononuclear cells (PBMCs) from healthy donors were incubated for 24 h with scorpion toxins rNATx or rSATx at 1, 10, or 20 µg per 200 µL reaction volume. Absolute numbers of total T-cells (CD3⁺) and T-cell subsets (CD4⁺, CD8⁺, and CD4⁺CD8⁺double-positive cells) were determined by flow cytometry using counting beads. Bar graphs represent mean±SD from triplicate measurements. rSATx exposure did not induce marked changes in total T-cell counts or subset distribution across the concentrations tested.

cell subpopulations, Th1 and Th2 play distinct roles: Th1 cells primarily produce IL-2, TNF-α and TNF-γ, and are involved in IgG production, while Th2 cells secrete IL-4, IL-5, IL-6, and IL-13, promoting the synthesis of IgE and IgA [69,70].

Neutralizing antibodies, particularly of the IgG2a subclasses, are critical in the development of effective scorpion antivenoms. IgG2a has been identified as a key immunoglobulin capable of conferring protection against lethal particles [71]. The venom of *T. serrulatus* has been reported to modulate the inflammatory response by increasing the levels of TNF-α, IL-6, and IL-1β [65,72,73]. Similarly, venom from *C. noxius* has shown *in vivo* effects on cytokine secretion, significantly elevating TNF-α, IFN-γ, and IL-6 levels for up to 21 days post-exposure [73,74].

Based on current literature, it has been reported that toxins from the genus *Centruroides* elicited a stronger immune response compared to other scorpions' toxins; however, the underlying mechanism responsible for this enhanced immunogenicity remains unclear. Addressing this phenomenon is of particular importance, as understanding why *Centruroides* toxins induce a more pronounced antibody response could provide valuable insights for the antivenom design and immunization strategies. In this context, the sustained elevation of IL-6 and other pro-inflammatory cytokines could partially explain why immunization with rNATx led to a more robust antibody response compared to rSATx. Although this hypothesis is supported by existing evidence, further studies are required to elucidate the molecular and immunological factors involved, highlighting its relevance for future research.

An important point to highlight in relation to the determination of the secondary structure of rNATx and rSATx toxins, the data obtained by CD when analyzing rSATx toxin showed a null presence of α-helix structure compared to rNATx toxin and the control rCssII whose β/α ratio was around 3:1, corresponding to the domains reported for sodium toxins (βαββ) present in scorpions [75]. The poor β-type structure in rSATx toxin could explain the production of antibodies whose ability to neutralize native toxins and whole venoms was not optimal; on the other hand, the data for rNATx toxin showed the presence of alpha and beta-type structures like the control rCssII toxin. Although both toxins were active in *in vivo* tests, the lack of alpha-like structure in rSATx could cause some conformations unfavorable for producing neutralizing antibodies. The data from CD presented in this communication are consistent with previously published data from our laboratory regarding the expression of scorpion toxins expressed in heterologous systems [24,34], and with the low immunogenicity of *Tityus* toxins in rabbits; in fact, the number of antibodies in a vial of an anti-*Tityus* venom from Brazil contains twice that of an anti-*Centruroides* from Mexico [44].

The ELISA assay showed that the venom of *C. limpidus* ($EC_{50}$: 0.0037) was the best recognized and the venom of *C. suffusus* ($EC_{50}$: 0.0049) the least recognized by the anti-rNATx; however, in the case of the anti-rSATx, the best recognized venom was that of *T. serrulatus* ($EC_{50}$: 1.04) and the worst was *T. festae* ($EC_{50}$: 1.39). *T. serrulatus* venom was the one that was completely neutralized, but not *T. festae* venom. The dose-response curves obtained in ELISA assays show significant differences in recognition by anti-rNATx compared to anti-rSATx antibodies. These findings are consistent with reported studies against toxins of the genus *Tityus*, which require a greater number of antibodies to be neutralized compared to antibodies raised against *Centruroides* toxins [34,57]. To neutralize *Centruroides* venoms, a maximum amount of 1.5 mg of anti-rNATx antibody was sufficient to neutralize $3LD_{50}$. However, in the case of *T. festae*, approximately 20 mg of anti-rSATx antibodies was necessary to achieve equivalent neutralization, representing a ~14-fold increase compared to the amount required for anti-rNATx.

## 5 Conclusions

The two recombinant consensus toxins used in this work were designed using the primary structure of lethal neurotoxins from *Centruroides* and *Tityus* species. After protein expression, rNATx and rSATx were accumulated in IB. Active isoforms were obtained after protein refolding under *in vitro* conditions. Antibodies generated against both recombinant toxins were assessed for their capacity to neutralize whole venoms or isolated toxins. Immunization with rNATx elicited a stronger and more consistent immune response than rSATx. Notably, antibodies raised against rNATx were able to neutralize the tested scorpion venoms, and it is anticipated that they may also exhibit broader neutralizing activity against additional *Centruroides* species. Concerning anti-rSATx antibodies, changes in the immunological scheme, including time, adjuvants, and protein doses, might be needed to improve antibody titers and antibody effectiveness. As for the cross-neutralization assays between anti-rNATx and anti-rSATx, they were unable to neutralize native toxins from the *Tityus* or *Centruroides* genera, respectively. Therefore, neutralization was directed only against the toxins and venoms that were the primary target.

The present findings provide a basis for the continued evaluation of recombinant scorpion toxins and their use as immunogens. While underscoring the need for future studies incorporating a larger sample size to strengthen and support the presented data. The development and optimization of novel antivenoms against medically important scorpions' species of the genera *Centruroides* and *Tityus* remain a critical public health priority, particularly in Latin America, where scorpion envenomation continues to impose a substantial burden. In this context, the integration of expanded statistical approaches will be essential for advancing the rational design of safer, more effective, and more accessible antivenoms.

## Acknowledgments

We are grateful to DVM Elizabeth Mata, BSc. Graciela Cabeza, Sergio González, and Oswaldo López from Unidad de Bioterio for handling the experimental animals. We also acknowledge D. Alejandro Fernández-Velasco for CD spectroscopy measurements. We additionally acknowledge to Dr. Paul Gaytán, M.C. Eugenio López-Bustos, and Q.I. Santiago Becerra de la Unidad de Síntesis y Secuenciación de ADN del Instituto de Bioecnología.

## Author contributions

**Conceptualization:** Samuel Cardoso-Arenas, Gerardo Corzo.

**Data curation:** Samuel Cardoso-Arenas, Miguel Angel Mejia-Sanchez, Ricardo Miranda-Blancas, Herlinda Clement, Lilu Corrales-García, Gerardo Pavel Espino-Solis, Gerardo Corzo.

**Investigation:** Samuel Cardoso-Arenas, Gerardo Corzo.

**Methodology:** Samuel Cardoso-Arenas, Miguel Angel Mejia-Sanchez, Herlinda Clement, Lilu Corrales-García, Ivan Arenas, Gerardo Pavel Espino-Solis, Gerardo Corzo.

**Project administration:** Gerardo Corzo.

**Resources:** Gerardo Pavel Espino-Solis, Hildaura Acosta, Marcos H. Salazar.

**Software:** Samuel Cardoso-Arenas, Ricardo Miranda-Blancas, Gerardo Corzo.

**Supervision:** Gerardo Corzo.

**Visualization:** Samuel Cardoso-Arenas, Gerardo Corzo.

**Writing – original draft:** Samuel Cardoso-Arenas, Gerardo Corzo.

**Writing – review & editing:** Samuel Cardoso-Arenas, Miguel Angel Mejia-Sanchez, Ricardo Miranda-Blancas, Gerardo Corzo.

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
