## [Decision Letter · Decision Letter 0]

6 Jan 2026

PNTD-D-25-01941

Immunogenicity of two representative American consensus scorpion neurotoxins from tropical scorpion species from the genera Tityus and Centruroides

Dear Dr. Corzo,

Thank you for submitting your manuscript to PLOS Neglected Tropical Diseases. After careful consideration, we feel that it has merit but does not fully meet PLOS Neglected Tropical Diseases's publication criteria as it currently stands. Therefore, we invite you to submit a revised version of the manuscript that addresses the points raised during the review process.

Please submit your revised manuscript within by Mar 07 2026 11:59PM. If you will need more time than this to complete your revisions, please reply to this message or contact the journal office at plosntds@plos.org. Please include the following items when submitting your revised manuscript:

We look forward to receiving your revised manuscript.

Kind regards,

Wuelton Monteiro, Ph.D.

Section Editor

Wuelton Monteiro

Section Editor

Shaden Kamhawi

co-Editor-in-Chief

Paul Brindley

co-Editor-in-Chief

**Journal Requirements:**

1) We do not publish any copyright or trademark symbols that usually accompany proprietary names, eg ©,  ®, or TM  (e.g. next to drug or reagent names). Therefore please remove all instances of trademark/copyright symbols throughout the text, including:

- TM on page: 8 and 9..

3) In the online submission form, you indicated that "Data available on request". All PLOS journals now require all data underlying the findings described in their manuscript to be freely available to other researchers, either

1. In a public repository

2. Within the manuscript itself

3. Uploaded as supplementary information.

4) Please amend your detailed Financial Disclosure statement. This is published with the article. It must therefore be completed in full sentences and contain the exact wording you wish to be published.

2) State what role the funders took in the study. If the funders had no role in your study, please state: "The funders had no role in study design, data collection and analysis, decision to publish, or preparation of the manuscript.".

**Reviewers' Comments:**

Reviewer's Responses to Questions

**Key Review Criteria Required for Acceptance?**

**Methods**

-Are the objectives of the study clearly articulated with a clear testable hypothesis stated?

-Is the study design appropriate to address the stated objectives?

-Is the population clearly described and appropriate for the hypothesis being tested?

-Is the sample size sufficient to ensure adequate power to address the hypothesis being tested?

-Were correct statistical analysis used to support conclusions?

-Are there concerns about ethical or regulatory requirements being met?

Reviewer #1: The study’s goals come across clearly, and the overall design makes sense for what the authors want to test. They do a nice job walking through how they built and tested the recombinant toxins. Still, the paper would benefit from stating the main hypothesis more directly it’s implied but not clearly written out. The experimental work looks solid, but the small number of animals limits how much we can read into the statistical comparisons. The data analysis is mostly descriptive, so adding even basic significance testing would strengthen the claims. It’s also good to see that animal ethics were handled properly and approved under institutional standards.

Reviewer #2: 1. Are the objectives of the study clearly articulated with a clear testable hypothesis stated?

Answer: Yes. The objectives are clearly stated: to design two consensus scorpion toxins (rNATx and rSATx) and evaluate their potential as immunogens to generate neutralizing antibodies against venoms from Centruroides and Tityus species. The implicit, testable hypothesis is that these designed consensus toxins will elicit a protective immune response that can neutralize the lethal effects of native venoms and toxins.

2. Is the study design appropriate to address the stated objectives?

Answer: Yes, broadly. The overall design—from in-silico design and recombinant expression, through protein refolding and characterization, to animal immunization and neutralization assays—is logical and appropriate to address the objectives. However, to fully validate the "consensus" concept, the design could be strengthened by including an assessment of cross-neutralization (e.g., testing anti-rNATx against Tityus venoms).

3. Is the population clearly described and appropriate for the hypothesis being tested?

Answer: Yes. The "populations" under study are clearly described:

Biological Toxins: The specific scorpion species (Centruroides and Tityus) and the individual venoms/toxins used are well-defined.

Experimental Animals: The species (mice, New Zealand rabbits), strains (CD-1), and weights are specified and are standard models for toxicology and immunology research in this field.

4. Is the sample size sufficient to ensure adequate power to address the hypothesis being tested?

Answer: No, there are concerns. The sample size for the critical in vivo toxicity and neutralization assays (n=3 mice per group) is a significant limitation. While common in preliminary studies, this small 'n' provides low statistical power, increases the susceptibility of results to individual animal variation, and makes it difficult to draw robust conclusions, especially for all-or-nothing endpoints like survival. A larger sample size is required to ensure adequate power. The authors should at a minimum discuss this as a study limitation.

5. Were correct statistical analysis used to support conclusions?

Answer: Cannot be determined. The manuscript states that GraphPad Prism was used but fails to specify the exact statistical tests employed for comparing groups (e.g., t-tests for EC50 comparisons, Log-rank test for survival curves). This omission makes it impossible to verify the validity of the statistical analysis. Furthermore, a key conclusion (rNATx has a "better immune response") is presented qualitatively without showing the supporting statistical comparison of the antibody titers.

6. Are there concerns about ethical or regulatory requirements being met?

Answer: No. The manuscript includes a clear statement regarding ethical approval from the institutional bioethics committee (CB/IBt/Project #285) and adherence to national animal care guidelines (NOM-062-ZOO-1999). This meets standard ethical and regulatory requirements for publication.

Reviewer #3: The experimental methodology is described in sufficient detail, particularly the strategy used to assemble the rNATx and rSATx genes prior to ligation into the ampicillin-resistant pQE30 vector and subsequent cloning into XL1-Blue competent cells. The recombinant toxins were characterized both in vitro and in vivo, and biological activity assays in mice were conducted in compliance with local ethical regulations.

**Results**

-Does the analysis presented match the analysis plan?

-Are the results clearly and completely presented?

-Are the figures (Tables, Images) of sufficient quality for clarity?

Reviewer #1: The results flow nicely from the methods and are easy to follow. The figures tell the story well especially the purification and ELISA data—but some could use clearer labels and slightly better image quality. The findings make it clear that rNATx generated stronger immune responses than rSATx, which is interesting and well supported. Still, a few numbers are missing that would make the results feel more complete like sample sizes, error bars, or p-values. Overall, the results are convincing, but a bit more detail in how variability was handled would make them stronger.

Reviewer #2: 1.Partially, but with significant gaps.

Match: The manuscript presents results for all the major experimental activities described in the Methods (e.g., protein expression, purification, CD spectroscopy, toxicity, ELISA, and neutralization assays). The analysis plan is generally followed.

Gaps: The most significant mismatch lies in the statistical analysis. The Methods mention the use of statistical software but do not pre-specify the tests. Consequently, key comparisons central to the main conclusion—specifically, the statistical comparison of the immunogenicity (EC50) between rNATx and rSATx—are not presented in the Results, despite the data being available in Figure 5. This crucial analysis is missing.

2.Clearly presented, but incompletely interpreted.

Clarity: The results are for the most part logically organized and described in the text. The figures and tables are referenced appropriately.

Completeness: The presentation is incomplete because the authors do not fully interpret or highlight their own data. As noted above, the quantitative difference in immunogenicity is not statistically analyzed or emphasized in the text. Furthermore, the neutralization results in Table 3 are not fully utilized; the potency is not calculated in the standard format (µg venom/mg IgG), which limits their clarity and comparability. The presentation would also be more complete if the additional bands in the Western Blots (Figures 1, 2) were briefly addressed.

3.Mostly yes, but with room for improvement.

Sufficient Quality: The SDS-PAGE gels, chromatograms, and CD spectra are of acceptable quality and support the conclusions drawn from them. The graphs in Figure 5 are clear.

Areas for Improvement:

Table 3 (Neutralization): The table contains the raw data but is not presented with optimal clarity. Reorganizing it to include the standard potency calculation and ensuring all fields are filled would greatly enhance its quality and usefulness.

Reviewer #3: The results rely primarily on early multiple sequence alignment (MSA) of amino acid sequences from Tityus and Centruroides species. While the authors assessed the neutralizing capacities of anti-rNATx and anti-rSATx antibodies against native toxins and whole venoms, using LD₅₀ values derived from previously published studies, the following major concerns should be addressed:

The analysis focuses exclusively on scorpion toxins known to induce lethal effects in mammals. However, the manuscript does not clearly explain how sequence motifs potentially involved in immune cross-reactivity were identified or prioritized. The rationale supporting the construction of the rNATx and rSATx consensus sequences is not sufficiently supported by detailed data analysis. As a result, it is difficult to follow the justification for the selection of individual amino acid residues at each position within the consensus sequences.

The neutralization experiments using rabbit anti-rNATx and anti-rSATx antibodies were performed against native toxins and whole venoms from different scorpion species. However, the LD₅₀ values employed were taken from previously published reports, without apparent reassessment of the LD₅₀ values for the specific venom batches used in this study. If the authors verified these values experimentally, this should be clearly stated and documented; otherwise, the reliance on historical LD₅₀ values should be justified.

Minor comments:

Section 3.1 / Table 1: Immunogenic motifs are not underlined, and references to previously published data are missing. This table would benefit from additional clarification and appropriate citation.

Section 3.5 / Table 2: Toxicity data are limited to a single tested dose (3 µg) of rNATx and rSATx. For increased scientific relevance, the estimation and reporting of LD₅₀ values by intracerebroventricular (i.c.v.) injection using the Spearman–Karber method are strongly recommended.

Section 3.6 / Figure 4: Further explanation is required to account for the observed differences in circular dichroism–derived secondary structures among rNATx, rSATx, and rCssII.

**Conclusions**

-Are the conclusions supported by the data presented?

-Are the limitations of analysis clearly described?

-Do the authors discuss how these data can be helpful to advance our understanding of the topic under study?

-Is public health relevance addressed?

Reviewer #1: The conclusions feel fair and grounded in the data. It’s clear that rNATx worked better and that both recombinant toxins have potential as immunogens. I appreciate that the authors discuss possible structural reasons for the difference, though they could talk more about the study’s limitations especially the small sample size and lack of statistical testing. It would also help to connect these findings more clearly to how they could improve real-world antivenom development in Latin America. The public health relevance is there; it just needs to be brought forward a bit more.

Reviewer #2: 1. Partially supported, but require qualification.

Supported: The core conclusion that both consensus toxins can be produced, used to generate specific antibodies, and that these antibodies can neutralize homologous venoms is strongly supported by the data (Figs. 3, 5; Table 3).

Requires Qualification: The conclusion that "rNATx was more immunogenic and had better antivenom efficacy" is only partially and qualitatively supported. The data suggest this (e.g., lower EC50 in ELISA, lower antibody dose needed for neutralization), but the claim lacks the direct statistical comparison of immunogenicity data to firmly support the "more" and "better" assertions. The conclusion should be rephrased to reflect the level of statistical support.

2. No. This is a significant weakness.

The manuscript lacks a dedicated section or paragraph discussing the study's limitations. Key limitations that should be acknowledged include:

The small sample size (n=3) in animal assays, which limits the statistical power and robustness of the conclusions.

The lack of cross-neutralization data, which is critical for validating the broad-spectrum potential of the consensus approach.

The absence of a direct statistical analysis comparing the immunogenicity of the two core immunogens.

The use of a non-physiological intracranial injection for toxicity testing, which may not fully recapitulate the pathophysiological events of a real envenoming.

3. Yes, adequately.

The authors effectively discuss the potential of their consensus strategy in the Introduction and Discussion. They contextualize their findings within existing literature on recombinant toxins and highlight how their approach could overcome the limitation of species-specific antivenoms. They specifically discuss how their work advances the field by providing a new method to generate broad-spectrum neutralizing antibodies using a single, designed immunogen.

4. Yes, clearly and effectively.

The public health relevance is a major strength of the manuscript. It is explicitly addressed in the Author Summary and the Introduction, which state that scorpion stings are a neglected health problem in tropical areas, causing significant morbidity and mortality. The study's direct aim is to develop a better intervention (improved antivenom) for this public health issue, making its relevance to the scope of PNTD very clear and compelling.

Reviewer #3: Overall, the discussion and conclusion sections would benefit from a more comprehensive and comparative analysis of the results. For example, although the authors report the induction of high antibody titers in rabbits, comparable to previously published studies, the corresponding immune response curves are not shown or discussed in sufficient detail.

**Editorial and Data Presentation Modifications?**

Reviewer #1: This paper is generally clear and well organized, but a few edits could make it smoother to read. The abstract should briefly state the hypothesis and summarize the key results with real numbers. A simple visual overview of the workflow would help readers grasp the study faster. In the figures, use consistent labeling and include error bars where possible. A few sentences could be polished for clarity for example, “got a better immune response” could be rephrased as “elicited a stronger immune response.” Reference formatting and data availability notes should also be double-checked for journal style. These are all light edits that would make a strong paper even better.

Reviewer #2: 1. Statistical Analysis and Reporting

Specify Statistical Tests: Please explicitly state the statistical tests used for all comparisons in the Methods section (e.g., "Differences in EC50 values were compared using an unpaired t-test," or "Survival curves were compared using the Log-rank test").

Quantify Key Conclusions: The central claim of differential immunogenicity between rNATx and rSATx should be directly supported by statistical analysis. We encourage a formal comparison (with p-value) of the anti-rNATx vs. anti-rSATx EC50 values when binding to their respective immunogens (from Figure 5). This will transform a qualitative observation into a robust, quantitative finding.

2. Data Presentation Clarity

Standardize Neutralization Potency: To align with field standards and improve comparability, please recalculate the data in Table 3 to present the neutralizing potency as "µg venom neutralized per mg of IgG" for all challenges. This provides a universal metric of antibody efficacy. The current "µg V / mg AV" column is incomplete and its calculation method is unclear.

Clarify Figure 5 Y-Axis: Consider labeling the Y-axis of Figure 5 more precisely as "Absorbance (450 nm)" to avoid any ambiguity.

Address Western Blot Annotations: A brief note in the figure legends for Figures 1 and 2 regarding the higher molecular weight bands (e.g., "Potential aggregates or non-specific binding are indicated by asterisks") would guide the reader and preempt questions.

3. Manuscript Structure and Language

Correct Section Numbering: The duplicated "2.2" subsections in the Materials and Methods should be renumbered to ensure a logical flow (e.g., 2.2, 2.3, 2.4, etc.).

Improve Language Flow: The manuscript requires thorough proofreading by a native English speaker or professional editing service to correct minor grammatical errors, improve sentence fluency, and ensure precise phrasing. Examples include:

"rNATx got a better immune response" -> "rNATx elicited a stronger immune response..."

"The antibodies produced were able to neutralize" -> "The resulting antibodies neutralized..."

Check for consistent spelling of Centruroides.

Streamline the Discussion:

The paragraph on cytokine responses (Lines 499-526) is informative but should be reframed. Please clarify that this is a hypothesis based on the literature for future investigation, as these measurements were not part of the current study.

The excellent point about the structural differences revealed by CD (Lines 527-541) should be more prominently featured as a primary explanation for the observed immunogenicity differences.

4. Minor Points

Figure 3 Legend: The legend mentions a gradient starting from 20% solvent B, while the main text (Line 204) mentions 0%. Please verify and correct this discrepancy.

Ethics Statement: The ethics statement is present but could be slightly expanded for completeness. Consider adding: "All efforts were made to minimize animal suffering and to use the minimum number of animals necessary to obtain robust data."

Reviewer #3: (No Response)

**Summary and General Comments**

Reviewer #1: This is a thoughtful and well-executed study that tackles a real public health problem using a smart molecular approach. The idea of consensus-designed toxins for broad antivenom coverage is creative and timely. The work is carefully done and clearly described, but it’s limited by small sample sizes and minimal statistical testing. Adding those analyses and tightening up the writing would make the paper ready for publication. Overall, it’s an interesting and promising contribution that deserves to move forward after revision.

Reviewer #2: Summary and General Comments

This manuscript describes the design, recombinant production, and immunological characterization of two consensus scorpion neurotoxins, rNATx (representing the Centruroides genus) and rSATx (representing the Tityus genus). The core objective is to evaluate their potential as single immunogens for generating broad-spectrum neutralizing antibodies against American buthid scorpion venoms.

Strengths and Novelty:

The principal strength of this work is its highly innovative concept. The use of computationally designed consensus toxins to circumvent the need for complex venom mixtures or multiple recombinant toxins is a significant and promising strategy in the antivenom field. The study is well-structured, proceeding logically from design to in vivo validation. The finding that rNATx appears to be a superior immunogen is interesting and could have practical implications for vaccine design.

General Execution and Scholarship:

The experimental execution is generally sound. The authors have successfully expressed, refolded, and characterized the recombinant proteins, demonstrating their structural and functional similarity to native toxins. The immunization and neutralization experiments are appropriately chosen to address the study's aims. The scholarship is adequate, with relevant literature cited to contextualize the work.

Weaknesses and Areas for Improvement:

The main weaknesses lie in data presentation, statistical rigor, and the depth of interpretation.

The small sample size (n=3) in animal assays is a notable limitation that weakens the statistical power of the conclusions.

A key analytical omission is the lack of a direct statistical comparison to substantiate the central claim of differential immunogenicity between rNATx and rSATx.

The discussion ventures into substantial speculation regarding cytokine-mediated mechanisms without supporting data from this study.

The presentation of neutralization data could be immediately improved by adopting the field-standard potency metric (µg venom/mg IgG).

Significance:

The work addresses an important public health issue—scorpion envenoming—and proposes a novel technological solution. If the potential of the consensus approach can be fully realized, it could significantly advance antivenom development. The results presented here provide a solid and encouraging proof-of-concept.

Recommended Modifications:

The revisions required are primarily focused on re-analysis and re-interpretation of existing data, along with enhanced methodological clarity. No new experiments are strictly mandated for acceptance, but addressing the following is essential:

Perform and report statistical comparisons of immunogenicity (EC50).

Adopt standard presentation for neutralization potency.

Revise the discussion to focus on data-driven explanations and explicitly acknowledge study limitations (e.g., sample size, lack of cross-neutralization data).

Correct minor editorial issues and improve language flow.

Publication Ethics:

I have no concerns regarding dual publication, research ethics, or publication ethics. The work appears to be original, and ethical approvals for animal research are appropriately documented.

Reviewer #3: The manuscript presents a study aimed at producing antibodies with broad-spectrum activity against Centruroides and Tityus scorpion venoms using two newly expressed recombinant toxins. Although the general approach of using recombinant or consensus toxins for antivenom development has been previously reported by multiple research groups worldwide, the authors clearly define their objective, which is based on newly designed consensus sequences. The evaluation of antibody cross-recognition by rabbit IgGs against venoms from Centruroides and Tityus species is of interest.

The potential relevance of the study lies in the design and cloning in Escherichia coli of two consensus sequences, rNATx and rSATx, with consideration given to critical residue positions and structural features relevant to immunogenicity. The recombinant toxin–expressing constructs were generated using customized overlapping and assembly PCR-based gene synthesis.

The experimental methodology is described in sufficient detail, particularly the strategy used to assemble the rNATx and rSATx genes prior to ligation into the ampicillin-resistant pQE30 vector and subsequent cloning into XL1-Blue competent cells. The recombinant toxins were characterized both in vitro and in vivo, and biological activity assays in mice were conducted in compliance with local ethical regulations.

The results rely primarily on early multiple sequence alignment (MSA) of amino acid sequences from Tityus and Centruroides species. While the authors assessed the neutralizing capacities of anti-rNATx and anti-rSATx antibodies against native toxins and whole venoms, using LD₅₀ values derived from previously published studies, the following major concerns should be addressed:

The analysis focuses exclusively on scorpion toxins known to induce lethal effects in mammals. However, the manuscript does not clearly explain how sequence motifs potentially involved in immune cross-reactivity were identified or prioritized. The rationale supporting the construction of the rNATx and rSATx consensus sequences is not sufficiently supported by detailed data analysis. As a result, it is difficult to follow the justification for the selection of individual amino acid residues at each position within the consensus sequences.

The neutralization experiments using rabbit anti-rNATx and anti-rSATx antibodies were performed against native toxins and whole venoms from different scorpion species. However, the LD₅₀ values employed were taken from previously published reports, without apparent reassessment of the LD₅₀ values for the specific venom batches used in this study. If the authors verified these values experimentally, this should be clearly stated and documented; otherwise, the reliance on historical LD₅₀ values should be justified.

Minor comments:

Section 3.1 / Table 1: Immunogenic motifs are not underlined, and references to previously published data are missing. This table would benefit from additional clarification and appropriate citation.

Section 3.5 / Table 2: Toxicity data are limited to a single tested dose (3 µg) of rNATx and rSATx. For increased scientific relevance, the estimation and reporting of LD₅₀ values by intracerebroventricular (i.c.v.) injection using the Spearman–Karber method are strongly recommended.

Section 3.6 / Figure 4: Further explanation is required to account for the observed differences in circular dichroism–derived secondary structures among rNATx, rSATx, and rCssII.

Overall, the discussion and conclusion sections would benefit from a more comprehensive and comparative analysis of the results. For example, although the authors report the induction of high antibody titers in rabbits, comparable to previously published studies, the corresponding immune response curves are not shown or discussed in sufficient detail.

PLOS authors have the option to publish the peer review history of their article (what does this mean? ). If published, this will include your full peer review and any attached files.

**Do you want your identity to be public for this peer review?** For information about this choice, including consent withdrawal, please see our Privacy Policy .

Reviewer #1: **Yes:** Dr. Arman Abdous DVM,MPH

Reviewer #2: No

Reviewer #3: **Yes:** Balkiss BOUHAOUALA-ZAHAR

**Figure resubmission:**
---

## [Editor Report · Decision Letter 1]

29 Jan 2026

Dear Dr. Corzo,

We are pleased to inform you that your manuscript 'Immunogenicity of two representative American consensus scorpion neurotoxins from the genera Tityus and Centruroides' has been provisionally accepted for publication in PLOS Neglected Tropical Diseases.

Best regards,

Wuelton Monteiro, Ph.D.

Section Editor

Wuelton Monteiro

Section Editor

Shaden Kamhawi

co-Editor-in-Chief

Paul Brindley

co-Editor-in-Chief

---

## [Editor Report · Acceptance letter]

Dear Dr. Corzo,

We are delighted to inform you that your manuscript, "Immunogenicity of two representative American consensus scorpion neurotoxins from the genera Tityus and Centruroides," has been formally accepted for publication in PLOS Neglected Tropical Diseases.

Best regards,

Shaden Kamhawi

co-Editor-in-Chief

Paul Brindley

co-Editor-in-Chief
